# An RFC4/Notch1 signaling feedback loop promotes NSCLC metastasis and stemness

Lei Liu[1,2,16], Tianyu Tao[1,16], Shihua Liu[3,16], Xia Yang[4], Xuwei Chen[1], Jiaer Liang[1], Ruohui Hong[1], Wenting Wang[1], Yi Yang[4], Xiaoyi Li[1], Youhong Zhang[5], Quanfeng Li[5], Shujun Liang[1], Haocheng Yu[6], Yun Wu[1], Xinyu Guo[7], Yan Lai[8], Xiaofan Ding[9], Hongyu Guan[10], Jueheng Wu [1], Xun Zhu[1], Jie Yuan[11], Jun Li[11], Shicheng Su [12], Mengfeng Li [1,5], Xiuyu Cai[13✉], Junchao Cai [14,15✉] & Han Tian [1✉]

Notch signaling represents a key mechanism mediating cancer metastasis and stemness. To understand how Notch signaling is overactivated to couple tumor metastasis and self-renewal in NSCLC cells, we performed the current study and showed that RFC4, a DNA replication factor amplified in more than 40% of NSCLC tissues, directly binds to the Notch1 intracellular domain (NICD1) to competitively abrogate CDK8/FBXW7-mediated degradation of NICD1. Moreover, *RFC4* is a functional transcriptional target gene of Notch1 signaling, forming a positive feedback loop between high RFC4 and NICD1 levels and sustained over-activation of Notch signaling, which not only leads to NSCLC tumorigenicity and metastasis but also confers NSCLC cell resistance to treatment with the clinically tested drug DAPT against NICD1 synthesis. Furthermore, together with our study, analysis of two public datasets involving more than 1500 NSCLC patients showed that *RFC4* gene amplification, and high RFC4 and NICD1 levels were tightly correlated with NSCLC metastasis, progression and poor patient prognosis. Therefore, our study characterizes the pivotal roles of the positive feedback loop between RFC4 and NICD1 in coupling NSCLC metastasis and stemness properties and suggests its therapeutic and diagnostic/prognostic potential for NSCLC therapy.

[1] Department of Microbiology, Zhongshan School of Medicine, Sun Yat-Sen University, Guangzhou, China. [2] Chongqing Key Laboratory of Molecular Oncology and Epigenetics, The First Affiliated Hospital of Chongqing Medical University, Chongqing, China. [3] State Key Laboratory of Oncology in South China, Collaborative Innovation Center for Cancer Medicine, Sun Yat-sen University Cancer Center, Guangzhou, China. [4] Department of Pharmacology, Zhongshan School of Medicine, Sun Yat-Sen University, Guangzhou, China. [5] Cancer Institute, Southern Medical University, Guangzhou, China. [6] Guangzhou No. 2 High School, Guangzhou, China. [7] The First School of Clinical Medicine, Southern Medical University, Guangzhou, China. [8] State Key Laboratory of Respiratory Diseases and Guangzhou Institute of Respiratory Health, First Affiliated Hospital of Guangzhou Medical University, Guangzhou, China. [9] Department of Surgery at the Sir YK Pao Centre for Cancer, The Chinese University of Hong Kong, Hong Kong, China. [10] Department of Endocrinology and Diabetes Center, The First Affiliated Hospital of Sun Yat-sen University, Guangzhou, China. [11] Department of Biochemistry, Zhongshan School of Medicine, Sun Yat-Sen University, Guangzhou, China. [12] Breast Tumor Center, Sun Yat-sen Memorial Hospital, Sun Yat-sen University, Guangzhou, China. [13] Department of General Internal Medicine, State Key Laboratory of Oncology in South China, Sun Yat-sen University Cancer Center, Guangzhou, China. [14] Department of Immunology, Sun Yat-sen University Zhongshan School of Medicine, Guangzhou, China. [15] Guangdong Engineering & Technology Research Center for Disease-Model Animals, Sun Yat-sen University, Guangzhou, China. [16]These authors contributed equally: Lei Liu, Tianyu Tao, Shihua Liu. ✉email: caixy@sysucc.org.cn; caijch3@mail.sysu.edu.cn; tianhan@mail2.sysu.edu.cn

Lung cancer is the most commonly diagnosed cancer type and the leading cause of cancer-related death worldwide[1]. Non-small-cell lung carcinoma (NSCLC) accounts for approximately 85% of all lung cancer cases, and primarily consists of adenocarcinoma, squamous cell carcinoma, and large cell carcinoma subtypes. It is estimated that nearly two-thirds of NSCLC patients show evidence of local or distant metastasis at the time of diagnosis, and only approximately 15% of patients with metastatic NSCLC survive 5 or more years after the diagnosis of metastases[2]. In addition to local metastasis in the lymphatic nodes (LNs) and contralateral lung, NSCLC cells often spread to various distant organs, commonly involving the bones, brain, and liver, either individually or simultaneously[3]. Despite the significant advancements in currently available therapies, the unsatisfying response rates of metastatic NSCLC patients to the initial anticancer treatment and the fairly high frequencies of tumor recurrence posttreatment remain to be the most serious challenge in the clinic[4,5]. It is therefore important to understand the causes of these clinical difficulties and identify effective therapeutic targets or prognostic biomarkers for the early identification of metastatic NSCLC.

It is well recognized that the poor prognosis and the difficulties in the clinical treatment of NSCLC are essentially ascribed to the metastasis and stemness properties acquired by some NSCLC cells[6]. The metastasis process involves serial cascade events[7]. Cancer cell stemness, which endows tumor cells with the self-renewal ability to develop visible tumors from a very small number of cells, contributes not only to tumorigenesis but also therapeutic resistance, tumor recurrence, and dissemination[8]. Notably, stemness and metastasis can be coupled during cancer development and progression[9]. For example, Mani et al.[10] reported that a cancer stem cell-like state can be induced by metastasis-inducing programming[10]. Tumor cells with a higher proportion of the self-renewal side-population usually propagate more rapidly[11,12]. However, the mechanistic foundation for the development of cancer stemness and metastasis properties, as well as the intrinsic links between the two properties, remains poorly understood.

At the molecular level, the properties of cancer stemness and metastasis can be regulated by single, common signaling pathways, among which the Notch pathway has been found to be widely and constitutively overactivated, especially in metastatic NSCLC[13,14]. Canonical Notch signaling is initiated by interactions between specific ligands, such as JAG family members on signal-sending cells and their receptors, such as Notch family members on signal-receiving cells; proteolysis of Notch at the extracellular juxtamembrane site by ADAM10 and then cleavage at the transmembrane domain of Notch by γ-secretase, then leads to the release of the intracellular domain of the Notch protein (NICD). NICD translocates into the nucleus and binds to the DNA-binding transcription factor RBP-Jκ, resulting in conversion of RBP-Jκ from a transcriptional repressor to a transcriptional activator, leading to consequent transcription of key downstream targets such as HES1, HES5, and HEY1, which are transcriptional repressors important for suppressing the transcription of differentiation-promoting genes[15–18]. It is important to note that several γ-secretase inhibitors had been tested in phase I/II clinical trials, including in patients with refractory metastatic or locally advanced NSCLC, which almost ended in therapeutic failures[19]. Thus, it is important to investigate how the Notch signaling is overactivated to couple metastasis and stemness properties.

Under physiological conditions, negative regulation is present to restrain overactivation of Notch signaling[20]. For example, CDK8 can be recruited to the NICD–RBP–Jκ complex to phosphorylate NICD, and the E3 ligase FBXW7 subsequently recognizes the phosphorylated NICD to cause its ubiquitin-dependent degradation in the nucleus[21,22]. In contrast, loss of negative regulation leading to NICD stabilization and high levels of NICD accumulation in the nucleus, representing a critical hallmark of overactivation of Notch signaling, have been widely implicated in tumor development and progression[23,24]. In this context, mutations in the heterodimer and PEST domains of Notch1, which confer ligand-independent cleavage of Notch1 and resistance to CDK8/FBXW7-mediated phosphorylation and degradation of NICD1, have been frequently identified in patients with acute T cell lymphoblastic leukemia (T-ALL)[25,26]. In addition, inactivating mutations in FBXW7, which lead to NICD1 stabilization and overactivated Notch signaling, are present in colorectal, gastric, pancreatic, endometrial, and blood cancers[27–29]. In NSCLC, however, genetic alterations in the important regulators of Notch signaling are rarely present[30–32]. Hence, how the negative regulation of Notch signaling is disrupted so that a high level of NICD is induced and maintained in NSCLC requires investigation.

In this study, we demonstrate that RFC4, one of several subunits of the replication factor C complex that functions in DNA replication and repair as a polymerase accessory protein, is frequently amplified in NSCLC and represents as a downstream transcriptional target of Notch1 signaling. Moreover, RFC4 tightly binds NICD1 to competitively abrogate CDK8/FBXW7-mediated phosphorylation and polyubiquitination to stabilize NICD1 proteins and overactivates Notch signaling in a positive feedback manner, conferring NSCLC cells both metastasis and stemness properties and resistance to γ-secretase inhibitor treatment, suggesting the therapeutic, diagnostic, and prognostic potential of the positive feedback loop consisting of high RFC4 and NICD1 levels.

## Results

**RFC4 is essential for Notch activation-induced metastasis and stemness of NSCLC.** To investigate whether Notch signaling plays a role during NSCLC progression, gene expression profiles of NSCLC patients with or without lymphatic nodes (LNs) metastasis downloaded from TCGA lung cancer datasets were analyzed using the gene set enrichment analysis (GSEA) method. Our results showed that Notch signaling was aberrantly activated in patients with LN metastasis (Fig. 1a). Moreover, in our cohort of 219 NSCLC patients, primary tumors exhibiting LN metastasis expressed higher protein levels of NICD1 than non-metastatic tumors, and high NICD1 protein levels correlated with short metastasis-free survival (Fig. 1b, c). The overexpression of NICD1 consistently conferred potent metastatic and stemness properties to NSCLC cells, as NICD1-overexpressing cells exhibited prominent systemic metastasis and lung metastases when they were injected intracardially and intravenously, respectively, and were able to form tumors when as few as $5 \times 10^3$ cells were subcutaneously inoculated (Fig. 1d–h). To identify the mediators crucial for NICD1-induced malignancy, NICD1-overexpressing A549 cells and corresponding vector-control cells were comparatively profiled for global gene expression (GSE137106). Among the genes significantly upregulated by NICD1 (Fig. 1i and Supplementary Data 1), silencing RFC4, one subunit of the replication factor C (RFC) complex, consisting of RFC1, RFC2, RFC3, RFC4, and RFC5, which function in DNA replication and repair as polymerase accessory proteins, not only significantly reversed NICD1-enhanced tumor cell proliferation and survival but also markedly reversed the effects of NICD1 on promoting the abilities to invade and to form tumor spheres, and on increasing the proportion of side-population (SP) cells and the expression of invasion- and stemness-promoting genes (Fig. 1j

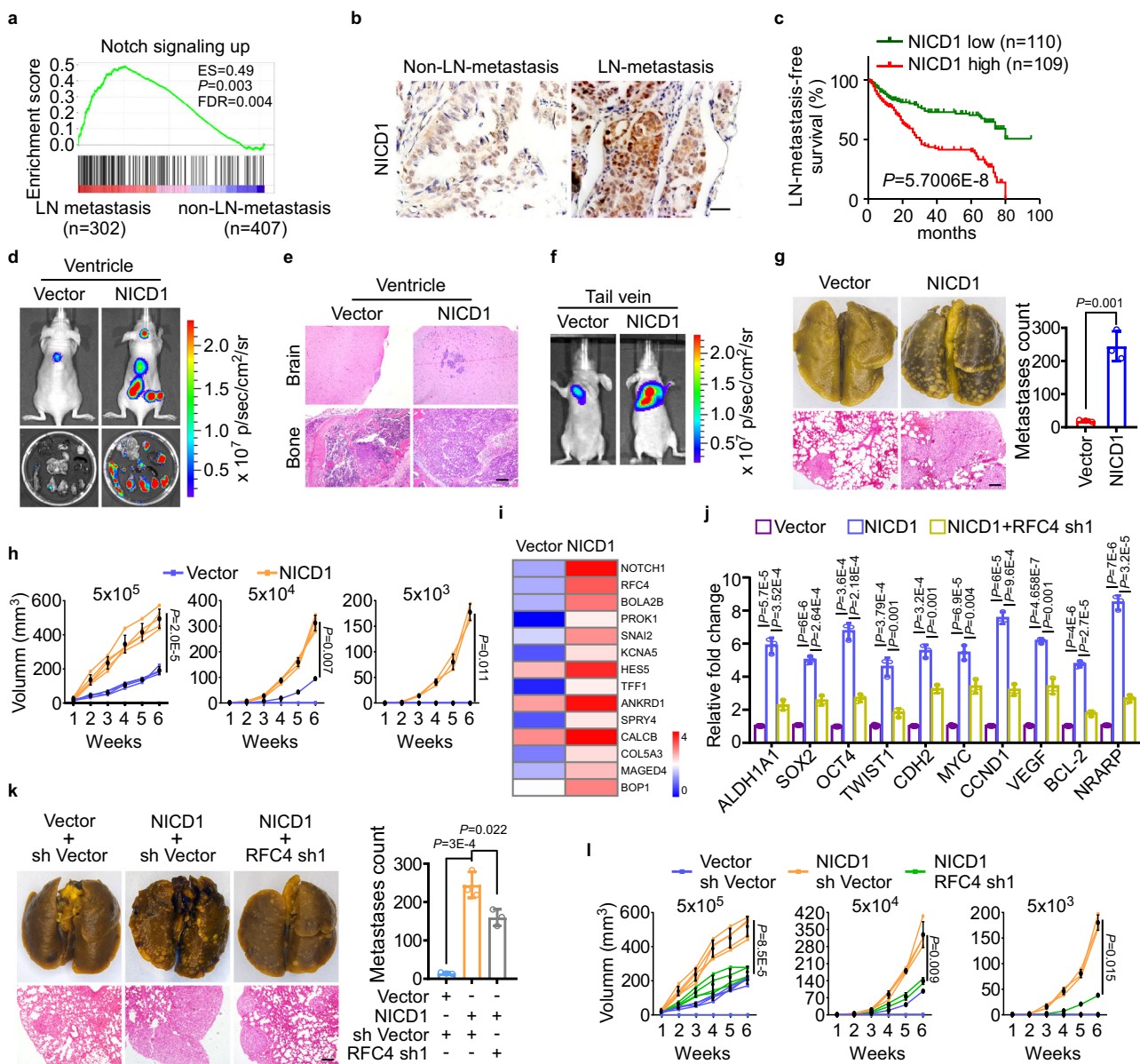

**Fig. 1 RFC4 is essential for Notch activation-induced metastasis and stemness of NSCLC. a** GSEA analysis of gene expression profiles of NSCLC patients with or without lymphatic node (LN) metastasis in the TCGA lung cancer datasets by Kolmogorov–Smirnov test. **b** IHC determined NICD1 expression in primary tumors of our cohort of 219 NSCLC patients with or without LN metastasis. Two representative cases are shown. Scale bar: 20 μm. **c** Kaplan–Meier analysis of LN metastasis-free survival of our 219 NSCLC patients by log-rank test, who were divided into low- or high- subgroups according to the median of NICD1 expression. **d**, **e** A549-luci-Vector or A549-luci-NICD1 cells were injected via cardiac ventricle into nude mice (n = 5 per group). Representative bioluminescent images of systemic metastasis and ex vivo organ metastases are shown (**d**), H&E histologically confirmed tumor cells in bone and brain tissue (**e**). Two representative cases are shown. Scale bar: 100 μm. **f**, **g** Nude mice were intravenously injected with A549-luci-Vector or A549-luci-NICD1 cells (n = 5 per group). Representative bioluminescent images (**f**), picric acid staining, H&E staining, and the numbers of metastatic foci of lung tissue are shown (**g**). Two representative cases are shown. Scale bar: 100 μm. **h** Growth curves of tumor xenografts of the indicated cells subcutaneously implanted with different cell numbers and tumor formation frequencies are shown. **i** Gene expression array analysis showed the most upregulated 10 genes induced by NICD1 overexpression. **j** The effect of silencing RFC4 on expression of the indicated mRNAs in NICD1-overexpressing cells. **k**, **l** The abilities of overexpressing NICD1 or together with RFC4-silenced A549 cells colonize in the lung or to generate tumor xenografts (n = 5 per group). Three representative cases are shown. Scale bar: 100 μm. Error bars represent the means ± SD derived from three independent experiments. Statistical analyses were performed by two-tailed unpaired Student's t-test (**g**), two-way ANOVA multiple comparison analysis (**h**, **j–l**). Source data are provided as a Source Data file.

and Supplementary Fig. 1a–g). Moreover, silencing RFC4 significantly compromised the invasive and self-renewal abilities of NICD1-overexpressing NSCLC cells, even when cell proliferation was inhibited by mitomycin C (a DNA synthesis inhibitor) treatment or cell death was induced by cisplatin treatment (Supplementary Fig. 1h, i). Consistently, A549-luci-NICD1 cells

silenced for RFC4 gave rise to weak bioluminescent metastatic signals when they were injected intravenously and generated small tumors only when more than $5 \times 10^4$ cells were subcutaneously inoculated, whereas scramble A549-luci-NICD1 cells presented potent metastatic and tumorigenic capacities (Fig. 1k, l and Supplementary Fig. 1j, k). Taken together, these data suggest

that NICD1-upregulated RFC4 might not only promote cell proliferation but also play a pivotal role in the metastasis and stemness properties induced by overactivated Notch signaling during NSCLC development and progression.

**RFC4 is a de novo direct transcriptional target of Notch1 signaling.** Interestingly, we further found that activating the Notch signaling by overexpressing NICD1 or by JAG1 treatment significantly increased RFC4 expression in A549 cells expressing low-level NICD1, whereas inhibiting the Notch1 activation by silencing Notch1 or by treatment with DAPT, a γ-secretase inhibitor, decreased RFC4 expression at both the protein and mRNA levels in H1975, LLC (Lewis lung carcinoma) and primarily cultured lung cancer cells (LC1) originating from a stage III lung adenocarcinoma (LUAD) patient's lung tumor, all of which express high levels of NICD1 and RFC4 (Fig. 2a, b and Supplementary Fig. 2a, b). Moreover, activating or inhibiting Notch1 signaling, respectively, promoted and abrogated the enrichment of activated histone H3K27ac in the promoter region of the *RFC4* gene in these NSCLC cells (Fig. 2c, d and Supplementary Fig. 2c). In parallel, silencing RBP-Jκ significantly inhibited RFC4 expression and reversed NICD1-induced upregulation of RFC4 expression, and inhibiting Notch1 activation similarly reversed the RBP-Jκ-induced increase in RFC4 levels (Fig. 2e, f and Supplementary Fig. 2d, e). We then investigated how RFC4 expression could be transcriptionally upregulated by NICD1, RBP-Jκ, and Notch signaling. By analyzing the promoter sequences of the *RFC4* gene using the ECR browser, a potential binding site for RBP-Jκ, the only transcription factor of canonical Notch signaling, was identified at an ENCODE H3K4Me1 site in the gene body, 1147 bp downstream of the transcription start site (Fig. 2g). Moreover, ChIP analysis revealed the binding of RBP-Jκ to the predicted site in the upstream promoter region of the *RFC4* gene in various NSCLC cells (Fig. 2h and Supplementary Fig. 2f). Additionally, when distinct, serial 500 bp DNA fragments containing the putative RBP-Jκ binding site with wild type or mutated sequences were separately cloned upstream of a luciferase reporter gene, only the luciferase activity of the reporter with wild-type sequences was significantly increased by NICD1 overexpression and inhibited by RBP-Jκ knockdown; silencing RBP-Jκ also markedly abrogated the above transcription-enhancing effect of NICD1 overexpression (Fig. 2i and Supplementary Fig. 2g), suggesting that the *RFC4* gene is indeed under direct transcriptional induction by RBP-Jκ and thus a downstream target gene of Notch signaling.

**RFC4 promotes NSCLC metastasis and stemness both in vitro and in vivo.** Consistent with the aforementioned finding of Notch signaling overactivation in NSCLC, our data showed that RFC4 expression was significantly upregulated in 81 out of 105 NSCLC tumor tissues as compared to their corresponding adjacent noncancerous lung tissues in the TCGA lung cancer datasets (fold change >2), as well as in 8 pairs of freshly resected NSCLC tissue specimens (Supplementary Fig. 3a–d). We then investigated the potential role of RFC4 in NSCLC development and progression. As shown in Fig. 3a; Supplementary Fig. 3e–h, overexpression of RFC4 in LUAD (A549) and lung squamous cell carcinoma (LUSC) (H1703) cell lines induced the expression of invasion- and stemness-promoting genes, potentiated the invasive and self-renewal abilities of these NSCLC cells to invade through the Matrigel and grow into more, larger-sized nonadherent cell spheres, and increased the proportions of SP cell fractions. In contrast, NSCLC cells silenced for endogenous RFC4 expression revealed greatly compromised invasive ability, along with weakened self-renewal ability compared with their corresponding

scramble-control cells (Fig. 3b and Supplementary Fig. 3e, i). In our in vivo studies, similar to the effect of NICD1, RFC4-overexpressing cells formed prominent metastases in various organ tissues, especially in the brain and bones, and exhibited remarkable lung colonization when they were injected intracardially and intravenously, respectively (Fig. 3c, d and Supplementary Fig. 3j, k). Impressively, mice either injected either intracardially or intravenously with RFC4-silenced NSCLC cells hardly presented metastatic bioluminescent signals (Fig. 3c, d and Supplementary Fig. 3j, k). In parallel, as few as $5 \times 10^3$ RFC4-overexpressing cells developed subcutaneous tumors, whereas more than $5 \times 10^4$ vector-control cells and $5 \times 10^5$ RFC4-silenced cells were required for tumor formation (Fig. 3e). Moreover, we used immunocompetent mice (C57BL/6N) to establish experimental metastasis and tumorigenicity models of RFC4-silenced and vector-control LLC cells. Consistently, silencing RFC4 significantly suppressed the ability of highly metastatic murine lung cancer cells to form lung metastases when injected intravenously or to develop subcutaneous tumors when injected with various cell numbers ranging from $5 \times 10^3$ to $5 \times 10^5$ in C57BL/6N mice (Fig. 3f, g and Supplementary Fig. 3l). Taken together, these data strongly demonstrate that high-level RFC4 promotes NSCLC metastasis and stemness properties as prominently as NICD1.

Notably, consistent with the role of RFC4 in DNA replication and repair, overexpression of RFC4 was significantly promoted, whereas knockdown of RFC4 markedly suppressed NSCLC cell proliferation, cell cycle progression, and resistance to cisplatin-induced cell apoptosis (Supplementary Fig. 4a–e). However, the promoting effects of RFC4 on the invasive and self-renewal abilities of NSCLC cells were not altered when cell proliferation was inhibited by mitomycin C treatment or cell death was induced by cisplatin treatment (Supplementary Fig. 4f, g). Additionally, the expression levels of RFC3 and RFC5, which bind RFC4 to form a core polymerase accessory complex, were rarely upregulated in NSCLC tissue compared to adjacent normal lung tissue (Supplementary Fig. 5a), and silencing RFC2 or RFC5 or silencing the essential DNA replication accessory gene PCNA reversed the promoting effects of RFC4 on NSCLC cell proliferation but failed to interfere with RFC4-induced Notch1 signaling activation or RFC4-potentiated tumor invasion and stemness in vitro or tumor metastasis and tumorigenicity in vivo (Supplementary Fig. 5b–i). These data suggest that the potent pro-metastatic and pro-self-renewal effects of RFC4 can be independent of its role in DNA replication and repair and its ability to enhance cell proliferation, cell cycle, and survival in NSCLC.

**RFC4 promotes NICD1 protein stability to form a positive feedback loop.** We then investigated the mechanisms underlying RFC4-induced aggressiveness of NSCLC cells. Interestingly, RFC4 overexpression largely increased, whereas silencing RFC4 decreased NICD1 protein levels without affecting the mRNA levels of Notch1 or the quantities of full-length Notch1 proteins both in various NSCLC cell lines and primary NSCLC cells (Fig. 4a and Supplementary Fig. 6a). Moreover, overexpressing or silencing RFC4 altered the amounts of NICD1 proteins in the nucleus but not in the cytoplasm (Fig. 4b), indicating that RFC4 might regulate NICD1 protein stability, which has been reported to involve ubiquitin-dependent degradation by a series of molecular events in the nucleus[33]. Indeed, silencing RFC4 caused a remarkable increase in K48-linked polyubiquitination of NICD1, and thus overexpressing RFC4 markedly prolonged, whereas silencing RFC4 greatly shortened the half-lives of NICD1 proteins in both various NSCLC cell lines and primary NSCLC cells

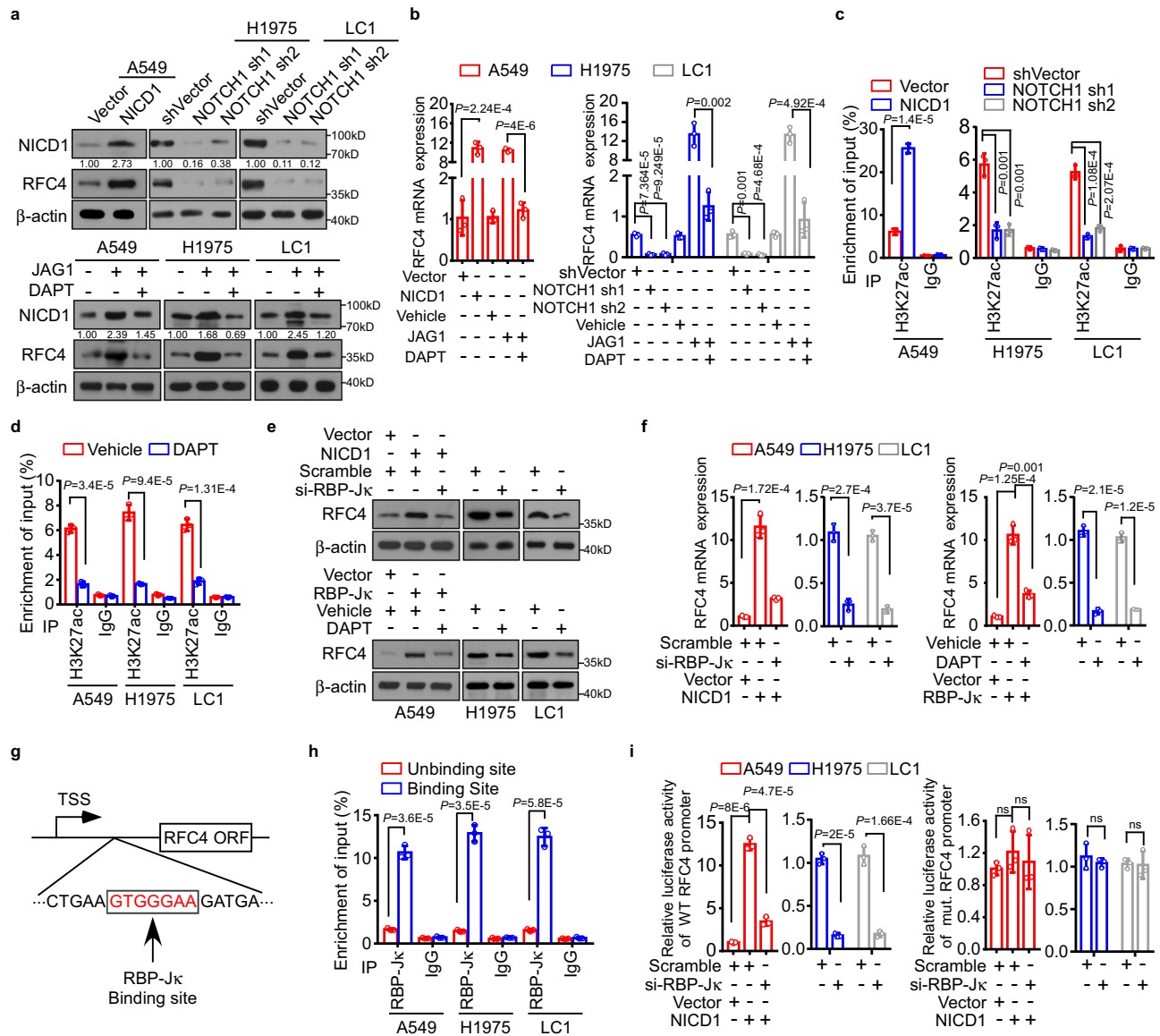

**Fig. 2 RFC4 is a de novo direct transcriptional target of Notch1 signaling. a**, **b** Effect of activating Notch signaling by overexpression of NICD1, JAG1, or inhibiting Notch signaling by treatment of a γ-secretase inhibitor DAPT on both protein and mRNA levels of RFC4 in A549, H1975, and LC1 cells. Representative images of three independent reproducible experiments are shown. **c**, **d** ChIP analysis following H3K27ac immunoprecipitation shows the interaction between H3K27ac and the promoter region of the RFC4 gene in response to overexpression of NICD1 or treatment of DAPT. IgG immunoprecipitation was used as a negative control. **e** The effect of silencing RBP-Jκ or overexpressing RBP-Jκ together with treatment of DAPT on protein levels of RFC4 in A549, H1975, and LC1 cells. Representative images of three independent reproducible experiments are shown. **f** The effect of silencing RBP-Jκ and overexpressing RBP-Jκ together with treatment of DAPT on mRNA levels of RFC4 in A549, H1975, and LC1 cells. **g** Schematic diagram of potential binding site for RBP-Jκ in the promoter region of RFC4. **h** ChIP enrichment assay shows binding of RBP-Jκ to the predicted binding site in the promoter region of RFC4 in stimulation of JAG1. IgG immunoprecipitation was used as a negative control. **i** The effects of overexpressing NICD1 together with RBP-Jκ depletion on luciferase activities of the reporter constructs spanning wild type or mutant predicted putative binding site for RBP-Jκ in the promoter region of RFC4. Data in panels **b**–**d**, **f**, **h** and **i** are presented as mean ± SD derived from three independent experiments. Two-way ANOVA multiple comparison analysis was used for statistical analysis. Source data are provided as a Source Data file.

(Fig. 4c, d and Supplementary Fig. 6b, c). Consequently, over-expressing RFC4 significantly promoted, whereas silencing RFC4 inhibited, the transcriptional activity of the Notch signaling and thus the expression of several canonical downstream genes in various NSCLC cells (Fig. 4e and Supplementary Fig. 6d). Similarly, silencing RFC4 reversed the activation of the Notch signaling pre-induced by NICD1 or JAG1 in NSCLC cells, and the Notch signaling in NSCLC cells pre-silenced with RFC4 became insensitive to stimulation of JAG1 overexpression in HUVECs (Supplementary Fig. 6e–g). Additionally, although RFC4

knockdown does not globally interfere with NICD-dependent transcription, silencing RFC4 reversed NICD1-induced expression of a set of Notch1 signaling downstream genes to various degrees (Supplementary Fig. 6h). These results suggest that NICD1-induced high-level expression of RFC4 promotes NICD1 protein stability to form a positive feedback loop, resulting in constitutive overactivation of the Notch signaling in NSCLC.

It is also important to note that, based on the RNA-seq data of tumor tissues of 971 NSCLC patients in the TCGA lung cancer datasets, the expression levels of RFC4 positively correlated with

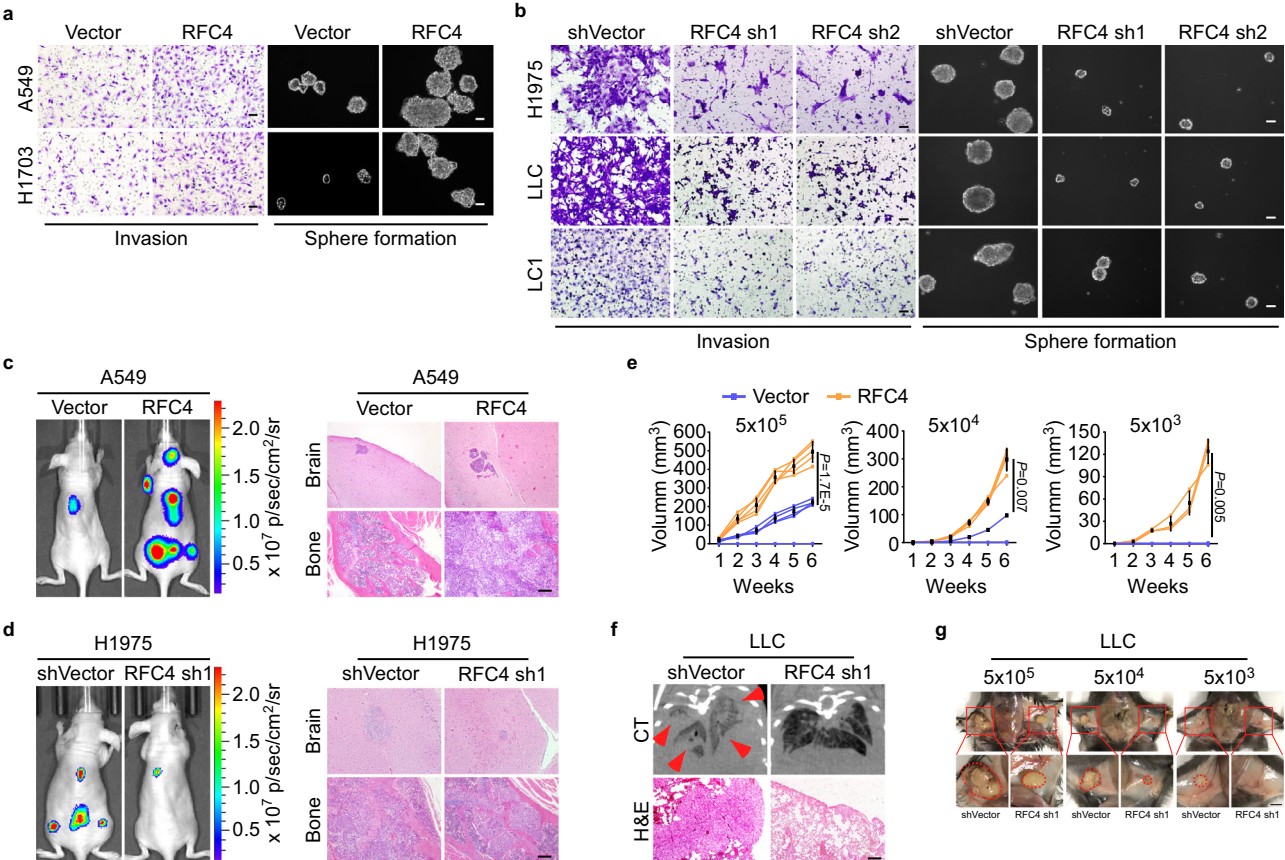

**Fig. 3 RFC4 promotes NSCLC metastasis and stemness both in vitro and in vivo. a** The effect of overexpressing RFC4 on the abilities of A549 cells to invade through matrigel or to form tumor spheres. Representative images of three independent reproducible experiments are shown. Scale bar: 50 μm. **b** The effect of silenced RFC4 on the abilities of H1975, LLC, and LC1 cells to invade through matrigel or to form tumor spheres. Representative images of three independent reproducible experiments are shown. Scale bar: 50 μm. **c** RFC4-overexpressing A549-luci cells were injected via cardiac ventricle into nude mice ($n = 5$ per group). Representative bioluminescent images of systemic metastasis and ex vivo organ metastases are shown (left panel). H&E histologically confirmed tumor cells in brain and bone tissue (right panel). Two representative cases are shown. Scale bar: 100 μm. **d** RFC4-slienced H1975-luci cells were injected via cardiac ventricle into nude mice ($n = 5$ per group). Representative bioluminescent images of systemic metastasis and ex vivo organ metastases are shown (left panel). H&E histologically confirmed tumor cells in brain and bone tissue (right panel). Two representative cases are shown. Scale bar: 100 μm. **e** Growth curves of tumor xenografts of RFC4-overexpressing A549 cells subcutaneously implanted with different cell numbers and tumor formation frequencies for indicated cell numbers are shown. **f** C57BL/6N mice ($n = 5$ per group) were intravenously injected with RFC4-silenced LLC cells. Micro-CT imaging and H&E staining of two representative cases' lung tissue are shown. Scale bar: 100 μm. **g** Tumor xenografts of RFC4-silenced LLC cells subcutaneously implanted with different cell numbers and tumor formation frequencies for indicated cell numbers are shown ($n = 5$ per group). Three representative cases are shown. Scale bar: 2 mm. Data in panel **e** were presented as mean ± SD derived from three independent experiments. Two-way ANOVA multiple comparison analysis was used for statistical analysis. Source data are provided as a Source Data file.

not only cell proliferation markers but also with overactivation of Notch1 signaling, especially with the expression levels of its several canonical downstream genes, such as HES1, HEY1, HEY2, and NRARP (Fig. 4f, g and Supplementary Fig. 6i). Consistent with these findings, we confirmed the positive correlation between RFC4 expression and the levels of NICD1 and HES1 in 219 NSCLC specimens collected in this study (Fig. 4h), suggesting a clinical relevance of the positive feedback loop consisting of activated Notch and high RFC4 levels during NSCLC development and progression.

**RFC4 binds to stabilize NICD1 by abrogating CDK8/FBXW7-induced degradation.** We next asked how RFC4 promotes NICD1 protein stability. Notably, using immunoprecipitation and mass spectrometry (MS) analysis, NICD1 was identified as a potential binding partner of RFC4, which was further validated by immunoprecipitation of endogenous or exogenous RFC4, purified recombinant RFC4 protein, and endogenous Notch1 or exogenous NICD1 (Fig. 5a and Supplementary Fig. 7a–c). In parallel, the

interaction between RFC4 and NICD1 was diminished when RFC4 and Notch1 were separately silenced (Fig. 5b). Moreover, a surface plasmon resonance (SPR) assay showed a direct interaction between purified proteins of RFC4 and NICD1 with a high binding affinity (Fig. 5c). We further found that RFC4 interacted with the C-terminal tail containing the PEST domain of NICD1 (Supplementary Fig. 7d). As serine phosphorylation in the PEST domain of NICD1 by CDK8 and subsequent recognition of phosphorylated NICD1 by its E3 ligase FBXW7 are essential for ubiquitin-dependent degradation of NICD1, we were prompted to examine whether RFC4 influences the interaction of NICD1 with CDK8 or FBXW7 and thus the degradation of NICD1. First, the SPR kinetic analysis showed that the binding affinity between RFC4 and NICD1 was approximately five folds higher than that between CDK8 and NICD1 (Fig. 5c). Second, the amounts of CDK8 or its binding partner Cyclin C were gradually impaired and even totally diminished in the pulled down proteins when HA-tagged NICD1 proteins were immunoprecipitated following the addition of purified RFC4 in a dose-dependent manner;

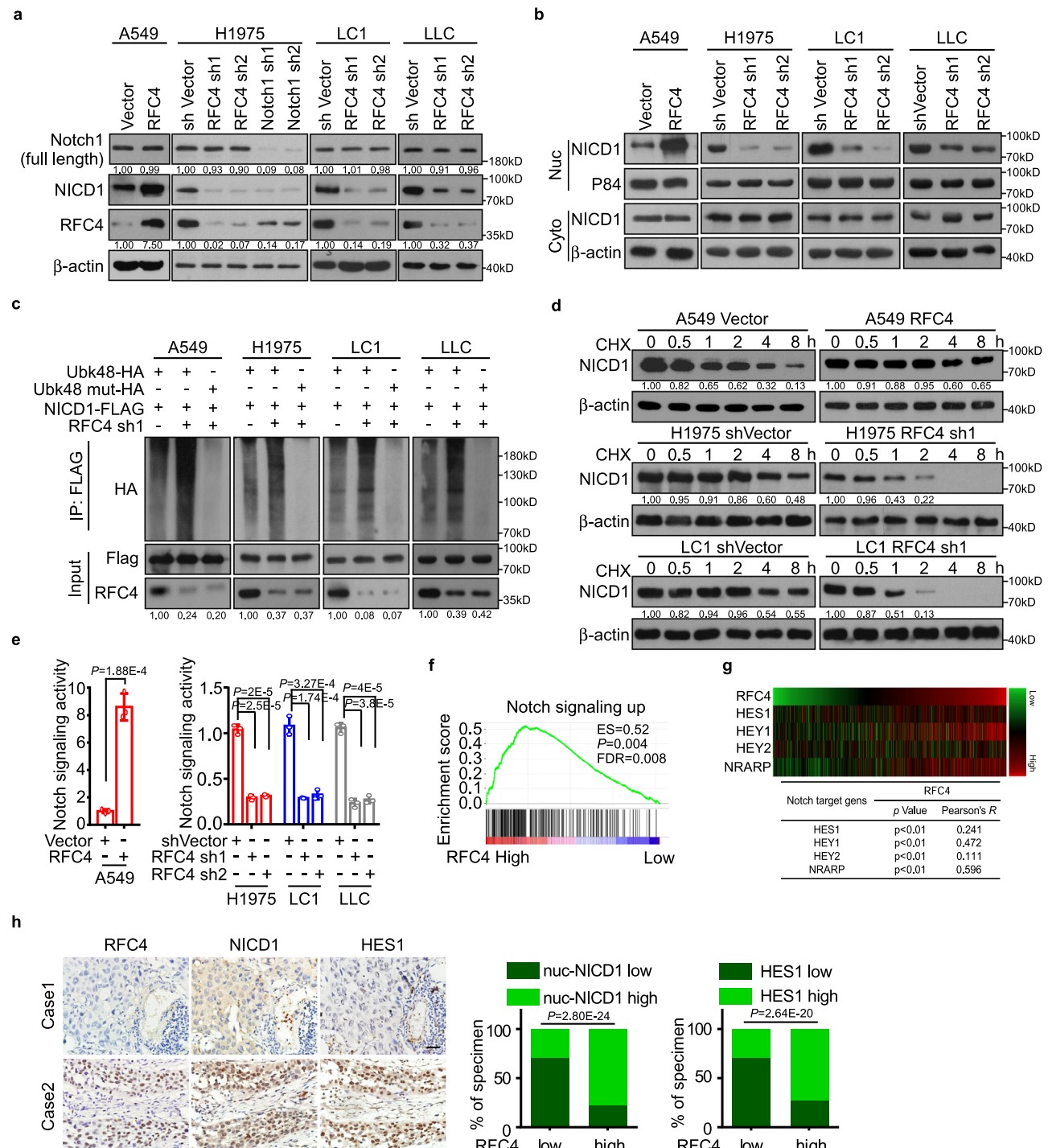

**Fig. 4 RFC4 promotes NICD1 protein stability to form a positive feedback loop. a**, **b** WB analysis of full-length Notch1, total NICD1, and subcellular distribution of NICD1 in indicated cells. **c** The effect of silencing RFC4 on the levels of K48-linked polyubiquitination of NICD1 was evaluated by immunoprecipitation of FLAG-tagged NICD1 in A549, H1975, LC1, and LLC cells. A dominant-negative mutant form of HA-tagged ubiquitin (UbK48R-HA) was used as a negative control. **d** The effect of overexpressing RFC4 on the half-lives of NICD1 in A549 cells and silencing RFC4 on the half-lives of NICD1 in H1975 and LC1 cells treated with cyclohexamide (CHX). **e** Dual-luciferase assays revealed Notch signaling activities in indicated cells. **f** GSEA analysis of the TCGA lung cancer datasets shows correlation between RFC4 and Notch signaling up signatures. **g** Pearson correlation analysis of the RNA-seq data of 971 NSCLC tissues in the TCGA lung cancer datasets showed the correlation between RFC4 expression and HES1, HEY1, HEY2, and NRARP levels. **h** Representative IHC staining images of two cases for RFC4, NICD1, and HES1 in the same set of consecutive NSCLC tissue slices, and correlations between RFC4 expression and levels of nuclear NICD1 and HES1 in 219 cases of NSCLC specimens are shown. Scale bar: 20 μm. Representative images of three independent reproducible experiments are shown (**a**–**d**). Data in panel **e**, **f**, **h** are presented as mean ± SD derived from three independent experiments. Two-way ANOVA multiple comparison analysis (**e**), two-tailed Student's *t*-test (**g**), and cross-tabulation with two-tailed Chi-square test (**h**) was used for statistical analysis. Source data are provided as a Source Data file.

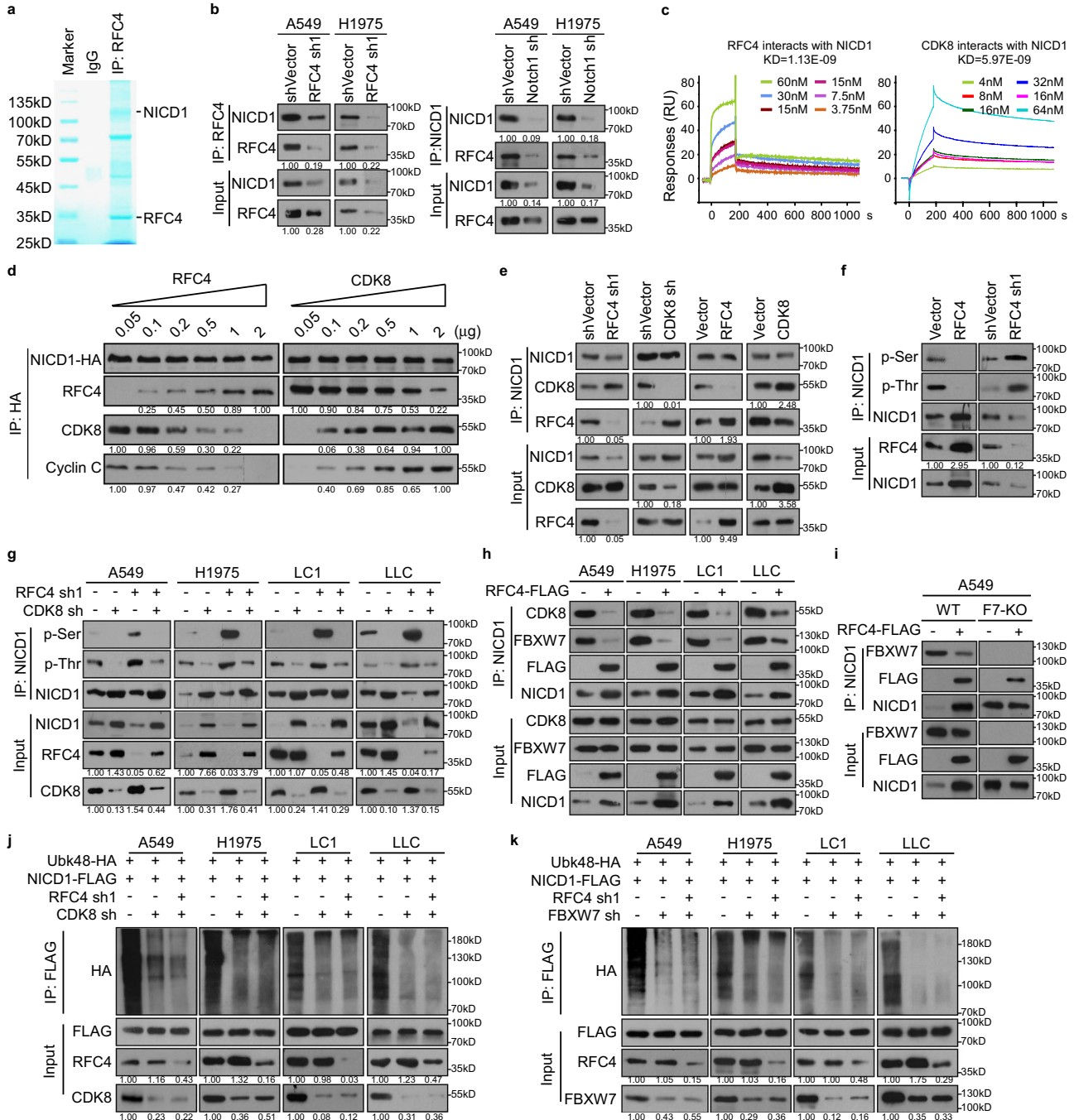

**Fig. 5 RFC4 binds to stabilize NICD1 by abrogating CDK8/FBXW7-induced degradation. a** MS peptide sequencing immunoprecipitated components, using anti-FLAG affinity purification, from lysates of A549 cells transfected with FLAG-tagged RFC4. **b** Immunoprecipitation assay revealing the interaction between RFC4 with NICD1. **c** SPR analysis measuring the affinity and kinetics of the interaction between RFC4 with NICD1 and CDK8 with NICD1. NICD1 was immobilized on a CM5 chip. **d** Lysates of 293FT cells transfected with NICD1-HA were immunoprecipitated by anti-HA affinity gel and incubated with purified RFC4 or CDK8 proteins, followed by addition of the indicated doses of purified CDK8, RFC4 proteins. The resultant incubates were analyzed by WB with the indicated antibodies. **e** The interaction between NICD1 and CDK8 or RFC4 in the presence or silencing of RFC4 or CDK8 was evaluated by immunoprecipitation of NICD1. **f, g** WB analysis of the effect of overexpressing or silencing RFC4 or together with CDK8 silencing on serine and threonine phosphorylation levels of immunoprecipitation pull-downed NICD1. **h** The interaction between NICD1 and CDK8 or FBXW7 in the presence or absence of RFC4-FLAG was evaluated by immunoprecipitation of Notch1. **i** The interaction between NICD1 and FBXW7 in the presence or absence of RFC4-FLAG was evaluated by immunoprecipitation of NICD1-HA in A549 WT and FBXW7 knockout cells. **j, k** The levels of K48-linked polyubiquitination of NICD1 were evaluated by immunoprecipitation of FLAG-tagged NICD1 in A549, H1975, LC1, and LLC cells with the indicated treatments. Source data are provided as a Source Data file, except Fig. 5c.

interestingly, although the binding of NICD1 to RFC4 could be impaired by the increasing amounts of purified CDK8 in a dose-dependent manner, it appears that large amounts of CDK8 proteins are required to significantly abrogate the interaction between NICD1 and RFC4 (Fig. 5d). Third, while silencing RFC4 mildly enhanced the interaction between NICD1 and CDK8, silencing CDK8 significantly enhanced the interaction between NICD1 and RFC4; in parallel, overexpressing RFC4 significantly abrogated the binding of NICD1 to CDK8, whereas CDK8 overexpression mildly impaired the binding of NICD1 to RFC4 (Fig. 5e). It is also important to note that from the TCGA lung cancer datasets, the mRNA levels of both CDK8 and Cyclin C were slightly increased in NSCLC tissue as compared to normal lung tissue; indeed, CDK8 protein levels were rarely upregulated in eight pairs of NSCLC tissues, whereas RFC4 protein levels were significantly upregulated (Supplementary Fig. 5a). These data suggest that both the higher binding affinity of RFC4 to NICD1 and the increased expression levels of RFC4 in NSCLC tissue make RFC4 more competitive than CDK8 in binding to NICD1.

As a result, RFC4 overexpression significantly decreased, whereas silencing RFC4 increased total serine and threonine phosphorylation of nuclear NICD1 proteins, and silencing CDK8 reversed the corresponding effects of silencing RFC4 on serine and threonine phosphorylation levels and total protein levels of nuclear NICD1, as well as on the transcriptional activity of Notch signaling (Fig. 5f, g and Supplementary Fig. 7e, f). Interestingly, overexpressing RFC4 still apparently reduced the total serine and threonine phosphorylation of NICD1(S2514A) or NICD1 (S2517A), whereas RFC4 hardly decreased total serine and threonine phosphorylation of NICD1(S2514A/S2517A); by contrast, RFC4 overexpression significantly reduced total serine phosphorylation of NICD1(T2512A), NICD1(T2542A), or NICD1(T2512A/2542A), and only reduced the total threonine phosphorylation of NICD1(T2512A) or NICD1(T2542A), but not of NICD1(T2512A/2542A) (Supplementary Fig. 7g). Notably, overexpressing RFC4 remarkably abrogated the binding of NICD1 to MEKK1 or GSK-3β (Supplementary Fig. 7h), which are potential kinases able to phosphorylate NICD1 on T2512 and T2542, respectively[34,35]. Based on these data, we hypothesize that the binding of RFC4 to the PEST domain of NICD1 might repel other NICD1-interactive proteins, including various NICD1 kinases, resulting in abrogation of NICD1 phosphorylation at multiple serine or threonine amino acids, such as S2514, S2517, T2512, and T2542, in the PEST domain. Furthermore, RFC4 overexpression also greatly abrogated the interaction between FBXW7 and NICD1 in vector-control A549 cells but not in FBXW7-depleted A549 cells (Fig. 5h, i). In parallel, silencing RFC4 failed to cause K48-linked polyubiquitination of NICD1 when FBXW7 or CDK8 was pre-silenced or depleted (Fig. 5j, k and Supplementary Fig. 7i). Taken together, these data suggest that RFC4 tightly binds with the C-terminal tail of NICD1 and competitively inhibits CDK8-mediated phosphorylation and FBXW7-mediated polyubiquitination, resulting in abrogated NICD1 degradation and stabilized NICD1 proteins.

**RFC4-induced stabilization of NICD1 promotes NSCLC aggressiveness and resists treatment with γ-secretase inhibitor.** To understand the significance of RFC4-induced stabilization of NICD1 in promoting NSCLC metastasis and stemness properties, an NICD1 mutant (NICD1mut) plasmid was employed that expresses an NICD1 protein constantly stabilized due to its ability to resist CDK8-mediated phosphorylation and thus leads to sustained activation of Notch signaling[18]. Indeed, silencing RFC4 markedly reversed the effect of overexpressing wild-type NICD1, but not that of NICD1mut, on NICD1 protein levels and Notch

signaling activation (Fig. 6a, b). Moreover, silencing RFC4 caused little, if any, alteration in the prominent inducing effect of stabilized NICD1 on the self-renewal and invasive abilities and SP increase of NSCLC cells (Supplementary Fig. 8a–c). Additionally, while NSCLC cells expressing stabilized mutant NICD1 developed excessive lung metastases when they were injected intravenously and were able to form detectable tumors even when $5 \times 10^3$ cells were subcutaneously inoculated, silencing RFC4 could not compromise the potent metastatic and tumorigenic abilities of these NICD1-mutant NSCLC cells (Fig. 6c and Supplementary Fig. 8d). Consistently, subcutaneous tumors formed by NICD1-mutant NSCLC cells presented significantly increased levels of RFC4, CCND1, and PCNA and a decreased proportion of TUNNEL-positive tumor cells as compared to those formed by vector-control cells, whereas subcutaneous tumors formed by RFC4-silenced NICD1-mutant NSCLC cells presented remarkable reductions in RFC4 protein levels and only marginally altered levels of NICD1, CCND1, or PCNA expression or apoptotic tumor cell proportion (Fig. 6d). Thus, these in vitro and in vivo data support our hypothesis that the RFC4-induced stabilization of NICD1 plays a pivotal role in promoting NSCLC metastasis and stemness properties.

Interestingly, in contrast to the remarkable suppressive effects of Notch1 silencing in RFC4-overexpressing NSCLC cells or DAPT treatment in Notch1-overexpressing NSCLC cells on their metastasis and stemness properties in vitro as well as in vivo, DAPT treatment slightly reversed these properties of RFC4- or NICD1-overexpressing cells (Fig. 6e–g and Supplementary Fig. 8e–g), suggesting that RFC4 promotes NSCLC metastasis and stemness properties in a Notch1 signaling-dependent manner, whereas NSCLC cells harboring high levels of RFC4 and NICD1, which constitute a positive feedback loop, are insensitive to γ-secretase inhibitor treatment.

**RFC4 is amplified in NSCLC and correlates with NSCLC progression.** To reveal whether RFC4-induced metastasis and stemness properties as identified above in NSCLC are clinically relevant, analysis of the RNA-seq profiles of the TCGA lung cancer datasets showed a significant correlation between the expression levels of RFC4 and molecular signatures related to cancer metastasis and stemness (Fig. 7a and Supplementary Fig. 9a). In our own cohort of 219 cases of NSCLC, primary tumors from patients bearing LN metastasis expressed higher levels of RFC4 than non-metastatic primary tumors (Fig. 7b). Moreover, in this cohort, RFC4 expression positively correlated with clinical staging and T-, N-, and M-classification of the included NSCLC patients (Supplementary Tables 1 and 2). Furthermore, patients with high RFC4 expression showed shorter overall and LN metastasis-free survival than those with low RFC4 expression and notably, patients with high levels of both RFC4 and nuclear NICD1 had significantly shorter LN metastasis-free survival times than those with high RFC4 or nuclear NICD1 alone (Fig. 7c and Supplementary Fig. 9b). Additionally, analyses of the online kmplot database (http://kmplot.com/analysis/) consisting of 1432 NSCLC patients and the MSKCC datasets consisting of 177 NSCLC patients showed that high RFC4 levels correlated with shorter overall survival time and progression-free as well as LN metastasis-free survival time (Supplementary Fig. 9c, d), further validating a potentially important role of RFC4 in the prognosis of NSCLC patients.

Consistent with previous reports showing amplification of chromosome region 3q27, where the *RFC4* gene is located in various types of cancers, including lung cancer[36–40], we found that 2.3% of 516 patients with LUAD and 40.3% of 501 patients with LUSC in the TCGA lung cancer datasets had *RFC4*

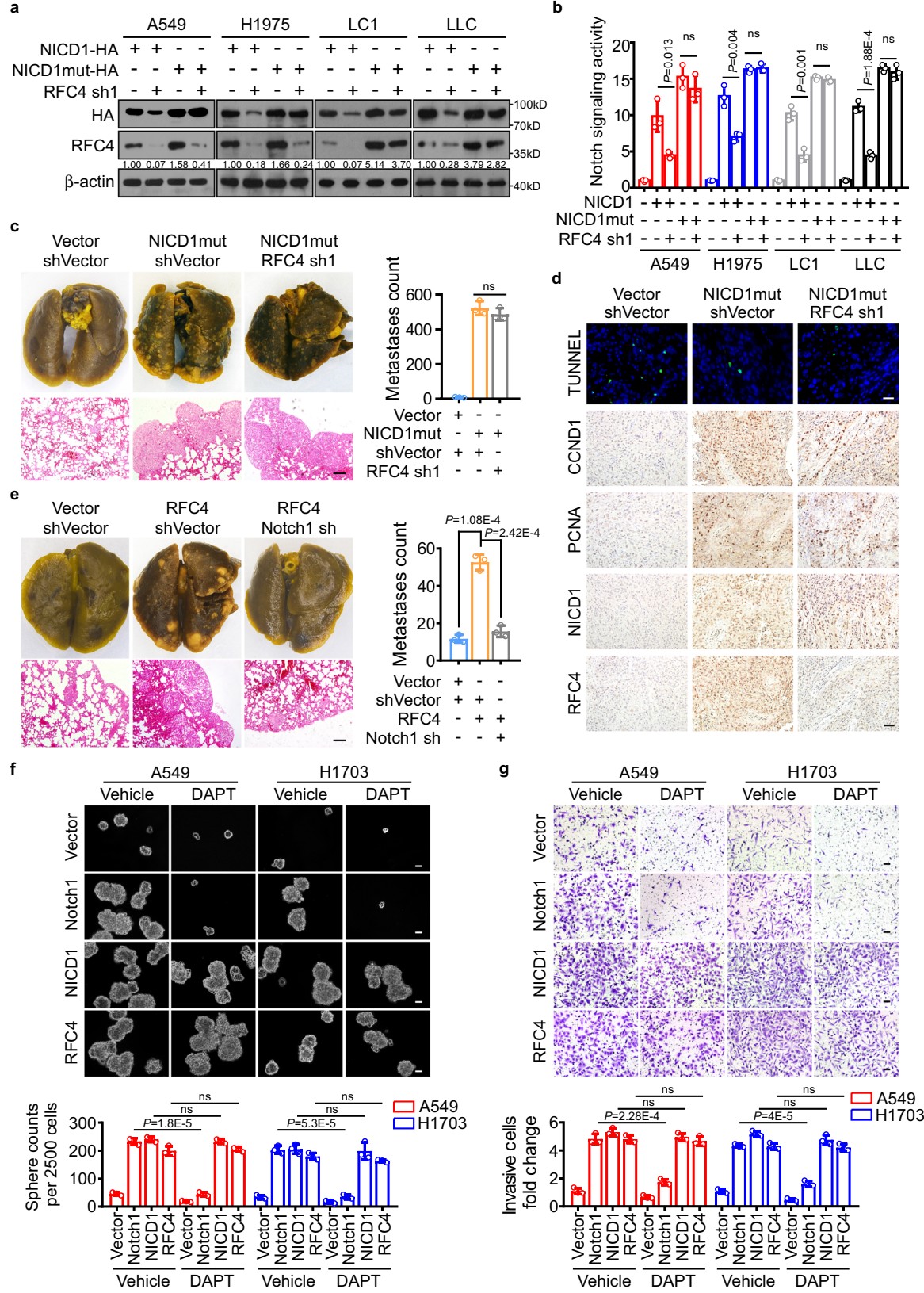

amplification in their lung tumors and that RFC4 mRNA levels were increased approximately 2.93- and 9.42-fold on average in LUAD and LUSC, respectively (Fig. 7d and Supplementary Fig. 9e). Moreover, the increase in genomic DNA levels of the *RFC4* gene was validated in 6 out of 10 pairs of NSCLC tissues as compared to the adjacent normal lung tissue, and RFC4 mRNA

expression indeed correlated with its DNA level (Fig. 7e and Supplementary Fig. 9f). Importantly, high copy numbers of *RFC4* DNA were mainly found in the primary tumors of NSCLC patients with local or distant metastasis (Fig. 7f, g). Furthermore, NSCLC patients with *RFC4* amplification or high *RFC4* DNA copy numbers in their lung tumors showed shorter overall or

**Fig. 6 RFC4-induced stabilization of NICD1 promotes NSCLC aggressiveness and resists treatment with γ-secretase inhibitor. a, b** The effect of silencing RFC4 on Notch signaling activities in A549, H1975, LC1, and LLC cells transfected with HA-tagged wild type or stabilized NICD1. Representative images of three independent reproducible experiments are shown. **c** A549 cells overexpressing stabilized NICD1 or together with RFC4 silencing were intravenously injected into nude mice ($n = 5$ per group). Representative images of picric acid staining, H&E staining, and the numbers of metastatic foci of indicated lung tissue are shown. Scale bar: 100 μm. **d** Representative TUNNEL and IHC staining images of CCND1, PCNA, NICD1, and RFC4 in tumor xenografts of the indicated cells ($n = 5$ per group). **e** The effect of silencing Notch1 in A549 cells overexpressing RFC4 on the metastatic and tumorigenic abilities. Scale bar: 100 μm. **f, g** The effect of DAPT treatment in A549 and H1703 cells overexpressing Notch1, wild-type NICD1, or RFC4 on their abilities to form tumor spheres or to invade through matrigel. Scale bar: 50 μm. Data in panel **b, c, e–g** are presented as mean ± SD derived from three independent experiments. Two-way ANOVA multiple comparison analysis was used for statistical analysis. Source data are provided as a Source Data file.

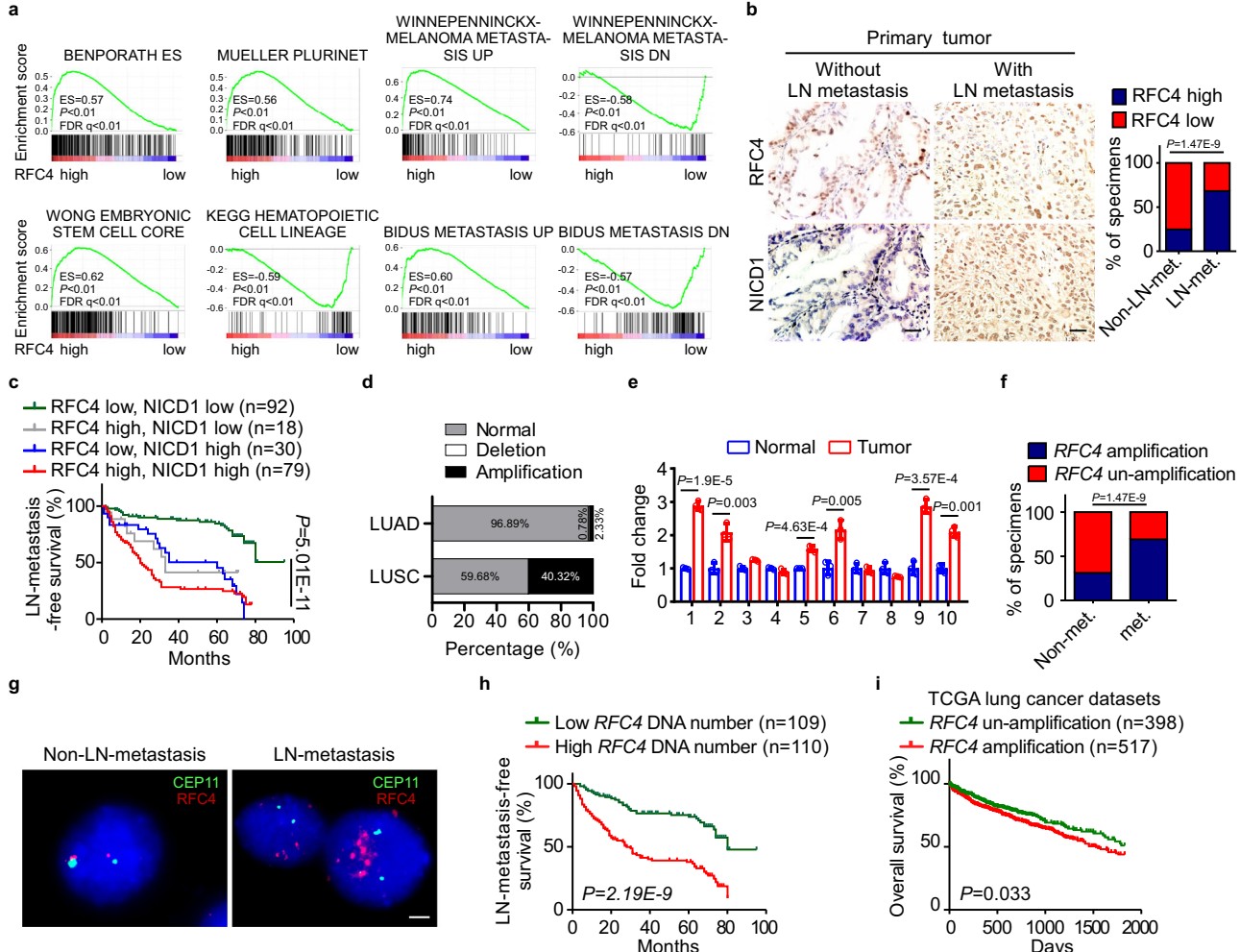

**Fig. 7 RFC4 is amplified in NSCLC and correlates with NSCLC progression. a** GSEA analysis of the TCGA lung cancer datasets by Kolmogorov–Smirnov test shows correlation between RFC4 and tumor metastasis- and stemness-associated signatures. **b** Images of IHC staining for RFC4 and NICD1 in the same set of consecutive tissue slices of primary tumors from 219 NSCLC patients with or without LN metastasis, and the percentages of specimens expressing low or high RFC4 from these NSCLC patients. Two representative cases are shown. Scale bar: 100 μm. **c** Kaplan–Meier analysis of the LN metastasis-free survival of our cohort of 219 NSCLC patients by log-rank test, who were divided into low- or high RFC4 and low- or high-NICD1 expression subgroups. The medians of RFC4 and NICD1 expression were used as the cut-off value. **d** The percentages of *RFC4* genetic alterations in LUAD and LUSC patients from the TCGA lung cancer datasets. **e** The *RFC4* DNA quantities in 10 pairs of NSCLC tissues and adjacent non-cancerous lung tissues. **f** The percentages of *RFC4* genetic alterations with or without metastasis from the TCGA lung cancer datasets, which was analyzed by cross-tabulation with two-tailed Chi-square test. **g** Images of FISH staining for *RFC4* genomic amplification status in primary tumors from NSCLC patients with or without LN metastasis. Two representative cases are shown. Scale bar: 10 μm. **h** Kaplan–Meier analysis of the correlation between *RFC4* DNA levels or amplification status and metastasis-free survival of NSCLC patients from our own cohort by Log-rank test. *RFC4* DNA level was used as the cut-off value. **i** Kaplan–Meier analysis of the correlation between *RFC4* amplification status and overall survival of NSCLC patients from TCGA lung cancer datasets by log-rank test. Data in panel **b, e** are presented as mean ± SD derived from three independent experiments. Two-way ANOVA multiple comparison analysis was used for statistical analysis. Source data are provided as a Source Data file.

metastasis-free survival than those without RFC4 amplification or with low *RFC4* DNA copy numbers (Fig. 7h, i), suggesting a role of *RFC4* gene amplification in the progression and possibly also the prognosis of NSCLC.

## Discussion

Biological and clinical evidence has demonstrated the important oncogenic roles of overactivated Notch signaling during the development and progression of various types of cancers, including NSCLC. In contrast to many other cancer types, the commonly found genetic alterations in genes such as *Notch1*, *JAG1*, and *FBXW7* central to NICD1 synthesis and Notch signaling are rarely present in NSCLC tumors[30–32]. Although loss of NUMB resulting from increased degradation of NUMB, an endocytic adaptor able to promote Notch1 degradation in the cytoplasm, has been reported in approximately 30% of LUAD tissues, it does not consistently or specifically cause Notch1 degradation; it has even been shown to exerts the opposite effects in lung squamous cell carcinoma[41,42]. Thus, the clinically relevant mechanisms underlying the high NICD1 level and Notch signaling overactivation in NSCLC remain largely unknown.

Our current study demonstrates that RFC4, which is amplified in the genomes of nearly 50% enrolled NSCLC cases, including both LUAD and squamous carcinoma subtypes, directly binds NICD1 with a high affinity to competitively abrogate CDK8-induced phosphorylation and FBXW7-induced ubiquitination-dependent degradation of NICD1, providing a biologically and clinically valid explanation for the observed increased NICD1 stability in the nucleus. Notably, while nuclear NICD1 displaces corepressors from the only transcription factor RBP-Jκ, which was originally identified as a transcriptional repressor, in canonical Notch signaling[43], it has not been elucidated how the repressive state of RBP-Jκ binding sites is disrupted, as NICD1 does not possess the ability to bind. In this context, as a DNA replication factor, RFC4 in the NICD1/RBP-Jκ complex might help to convert the responsive elements of the target genes from repressive to transcriptionally active for RBP-Jκ activation. Moreover, our study also identifies *RFC4* as a downstream target gene of Notch signaling based on our identification of direct RFC4 transcription by RBP-Jκ, suggesting the presence of a positive feedback loop consisting of high levels of both RFC4 and NICD1. Of note, although the Notch signaling is active in stem cells and highly malignant cells, most Notch target genes previously identified are differentiation- or proliferation-regulating genes[44,45]. Based on our findings of a robust capacity role of RFC4 in inducing stemness and metastasis properties in NSCLC, as well as a positive correlation of RFC4 with the stemness- and metastasis-promoting signature, as shown by the analysis of a large number of RNA-seq profiles of TCGA lung cancer datasets, it is highly likely that *RFC4* represents a functional Notch target gene. In this context, we propose that a positive feedback loop initiated by *RFC4* gene amplification causes amplified and sustained overactivation of Notch signaling and facilitates transcriptional activation of a set of metastasis- and cancer stemness-promoting genes in NSCLC. However, as RCF4 expression also significantly correlates with cell proliferation markers in human NSCLC tissue, the oncogenic effects of RFC4 on cell proliferation, tumorigenicity and tumor metastasis could together contribute to the worse outcomes of NSCLC patient prognosis.

It is well acknowledged that NSCLC can rapidly develop distant metastases and resistance to drug therapies. Such a characteristic can be attributed, at least partly, to the metastatic and stemness properties acquired by fractions of NSCLC cells. Consistent with the previously recognized notion that Notch signaling activates directly or indirectly activates the expression of stemness- and metastasis-promoting genes[46,47], our current study suggests that NSCLC cells with high levels of RFC4 and nuclear NICD1, which might constitute a positive signaling feedback loop, show potent metastasis and stemness properties and therefore might represent reasonable targets for the treatment of NSCLC metastasis and drug resistance. It is also notable that NICD1- or RFC4-overexpressing NSCLC cells are insensitive to γ-secretase inhibitor treatment, and several γ-secretase inhibitors already tested in phase I/II clinical trials for various types of cancers, including in NSCLC patients with refractory metastatic or locally advanced diseases, failed due to adverse effects and suboptimal efficacies, largely attributable to the complexity of Notch inhibition and possibly alternative oncogenic signaling contributed by other pathways[48]. Interestingly, our present study suggests that the positive feedback loop initiated by *RFC4* gene amplification constitutively generates high NICD1 levels independent of γ-secretase-mediated cleavage. Additionally, it has been suggested that Notch pathway mutations are not the most suitable biomarkers for predicting NSCLC response to γ-secretase inhibitors[48,49]. Hence, future in-depth investigation is warranted to elucidate whether the observed insensitivity of NSCLC cells to γ-secretase inhibitors is due to harboring a positive feedback loop in these cells; therefore, NSCLC patients with low RFC4 levels or without *RFC4* amplification should be selected for γ-secretase inhibitor therapy. Moreover, it may be worth developing RFC4-targeted therapeutic strategies against NSCLC.

Additionally, in many types of squamous cancers, such as squamous cell carcinoma of the head and neck, skin, oral cavity, and esophagus, 10–20% of tumor cases harbor inactivating mutations in the *Notch1* gene, and many studies suggest the tumor-suppressive roles of Notch1 in these squamous cancers[50–52]. In LUSC, approximately 5% of patient samples harbor inactivating mutations in the *Notch1* gene[53]. However, both the oncogenic and tumor-suppressive roles of Notch1 have been reported in LUSC[54,55]. Interestingly, as analyzed from the TCGA lung cancer datasets and KM plot database, high mRNA levels of Notch1 significantly correlate with poor overall survival and disease progression of LUSC patients (Supplementary Fig. 3). Our current study also employs a LUSC cell line (H1703) to prove the pro-invasive and pro-tumorigenic effects of RFC4 and RFC4-directed stabilization of NICD1 proteins and thus activation of Notch signaling, suggesting that RFC4-caused overactivation of Notch1 signaling plays important tumor-promoting roles during LUSC progression. On the other hand, consistent with the notion that amplification of the 3q chromosome, where the *RFC4* gene is located, is mostly seen in squamous cancers, the mRNA levels of RFC4 are averagely increased approximately 9.42-fold in LUSC, approximately 40.3% of which have *RFC4* gene amplification in the TCGA lung cancer datasets; by contrast, RFC4 mRNA levels are increased approximately threefold on average in LUAD, and approximately 2.33% of LUAD samples have *RFC4* gene amplification in the TCGA lung cancer datasets. Our study suggests that both the transcriptional upregulation of RFC4 by activated Notch1 signaling and *RFC4* amplification should contribute to high levels of RFC4 to varying degrees in LUAD and LUSC, both of which could utilize the pivotal roles of the positive feedback loop between RFC4 and NICD1 in coupling NSCLC metastasis and stemness properties. Additionally, we found that high levels of RFC4, as well as *RFC4* gene amplification, were more frequently found in primary NSCLC tumors from patients bearing LN metastasis and correlate with short overall survival and metastasis-free and progression-free survival time, indicating potentially promising diagnostic and prognostic values of RFC4, especially for metastatic NSCLC patients. Notably, other cancer types, such as esophageal and ovarian cancers, in which aberrantly activated Notch1 signaling plays important roles, also have distinct high proportions of *RFC4* gene amplification, and high

RFC4 levels significantly correlates with poor prognosis of patients with these cancers, indicating that the oncogenic effects of RFC4 are not limited to NSCLC.

## Methods

**Clinical specimens.** All clinical tissue specimens used in this study were obtained from and histopathologically diagnosed at the SYSU Cancer Center. The histological characterization and clinicopathologic staging of tumor samples were determined following the standard provided in the current Union for International Cancer Control (UICC) Tumor-Node-Metastasis (TNM) classification. Adjacent non-cancerous lung tissue specimens were collected from a standard distance (3 cm) from the margin of resected neoplastic tissues of NSCLC patients. For the use of these clinical materials for research purposes, prior patients' consents and approval from the SYSU School of Medicine Institutional Research Ethics Committee were obtained.

**Cell culture.** NSCLC cell lines, including A549, H1975, H1703, mouse LLC (LL/2) cell, non-cancerous HEK293FT (293FT), and human umbilical vein endothelial (HUVEC) cells were obtained from the Cell Bank of Shanghai Institutes of Biological Sciences (Shanghai, China) or ATCC, and cultured in Dulbecco's modified Eagle's medium (DMEM) (GIBCO) medium supplemented with 10% fetal bovine serum and 1% penicillin/streptomycin (penicillin 100 U/ml and streptomycin 10 µg/ml) or LSGS-supplemented medium 200PRF (for HUVEC, GIBCO). Primary normal lung epithelial and primarily cultured stage III LUAD cell (LC1) were cultured in Defined Keratinocyte SFM (GIBCO) supplemented with L-glutamine, EGF (20 ng/ml), basic-FGF (10 ng/ml), 2% B27, penicillin/streptomycin, and amphotericin B (0.25 mg/ml)[56,57]. All cell lines were authenticated by short tandem repeat (STR) fingerprinting at Medicine Laboratory of Forensic Medicine Department of Sun Yat-Sen University (SYSU) (Guangzhou, China), and were tested to be free of mycoplasma contamination.

**Plasmids, virus production, and transfection.** The open reading frames of RFC4, NICD1, and NICD1 mutant resistant to CDK8/FBXW7-mediated degradation were generated by PCR amplification and subcloned into the pSin-EF2 lentiviral vectors (Addgene) with different antibiotic resistance genes and various deletion mutants of HA-tagged NICD1 and FLAG-tagged RFC4 or NICD1, as well as HA-tagged ubiquitin, were subcloned into a pcDNA 3.1 vector. Plasmids of pCMV6-Entry-Myc-Notch1, pCMV6-Entry-HA-NICD1mut (P2515R), pCMV3-JAG1-Flag, and the Notch signaling reporter were purchased from OriGene, Upstate Biotechnology or Sino Biological Inc. (Beijing, China). Recombinant His-tagged RFC4 was expressed using pET-19b vector. For depletion of RFC4, two human shRNA sequences were cloned into pSuper-retro-puro or pSuper-retro-neo retroviral vectors. All siRNA oligonucleotides were purchased from Ribo (Guangzhou, China). Stable cell lines were generated via retroviral or lentiviral infection and selected with appropriate antibiotics for 10–14 days. Transfection of plasmids or RNA oligonucleotides was performed using Lipofectamine 3000 reagent (Invitrogen) for luciferase reporter assays and molecular assays. Oligos used for knockdown or knockout genes are listed in Supplementary Table 3.

**Western blotting (WB) analysis.** WB analysis was performed according to the protocol of a standard method. Primary antibodies used for WB analysis were anti-RFC4 (Abcam, Cambridge, MA, ab156780, 1:1000), anti-Notch1 (Cell Signaling, Danvers, MA, 3608, 1:500), anti-NICD1 (Abcam, Cambridge, MA, ab8925, 1:500), anti-p-Ser (Abcam, Cambridge, MA, ab9332, 1:500), anti-p-Thr (Cell Signaling, Danvers, MA, 9381s, 1:500), anti-CDK8 (Abcam, Cambridge, MA, ab224828, 1:1000), anti-FBXW7 (Abcam, Cambridge, MA, ab109617, 1:1000), anti-cyclin C (Abcam, Cambridge, MA, ab85927, 1:1000), anti-RFC2 (Abcam, Cambridge, MA, ab174271, 1:1000), anti-RFC5 (Abcam, Cambridge, MA, ab79871, 1:200), anti-GSK-3β (Abcam, Cambridge, MA, ab32391, 1:1000), anti-MEKK1(Abcam, Cambridge, MA, ab212601, 1:1000), anti-FLAG (Sigma, Saint Louis, MO, USA, F7425, 1:2000), anti-HA (Sigma, Saint Louis, MO, USA, H6908, 1:2000), anti-MYC (Cell Signaling, Danvers, MA, 2278, 1:2000), anti-His antibodies (Abcam, Cambridge, MA, ab9108, 1:2000). Blotted membranes were stripped and re-blotted with anti-p84 (Abcam, Cambridge, MA, ab131268, 1:2000) and anti-β-actin (Sigma, Saint Louis, MO, USA, A2228 1:2000), used as loading controls.

**RNA extraction and real-time PCR.** Total RNA from cultured cells and frozen surgical NSCLC tissues was isolated with TRIzol reagent (Invitrogen) as instructed. cDNA was synthesized from 2 µg of total RNA with random primers with the use of the Gene Expression Assays (Promega) and analyzed with Biorad CFX Manager 3.1 software. Expression of mRNAs was assessed based on the threshold cycle (Ct), and relative expression levels were calculated as $2^{-[(Ct\,of\,mRNA)-(Ct\,of\,GAPDH)]}$ after normalization to GAPDH expression. Experiments were performed at least three times, with triplicate replicates. Sense and antisense primers used for quantitative reverse transcriptase PCR are listed Supplementary Table 4.

**GSEA and microarray data deposition.** The Cancer Genome Atlas (TCGA) and the Memorial Sloan Kettering Cancer Center (MSKCC) NSCLC datasets were downloaded, respectively, to identify the association of RFC4 with stemness-related and metastasis-related gene signatures using the GSEA software, or to analyze the prognostic significance of RFC4 and NICD1. In addition, total RNAs from A549-Vector and A549-NICD1 cells were collected for mRNA-sequencing analysis by Berry Genomics (Beijing, China) following the standard protocol and sequencing data have been deposited in GEO database with accession number GSE137106. Bioinformatics analysis and visual heatmaps were performed with the MeV 4.4 program.

**Immunoprecipitation and protein purification.** Lysates were prepared from $3 \times 10^{7}$ 293FT cells transfected with Flag-, or HA-tagged RFC4 or full-length or truncated NICD1 in an NP-40-containing lysis buffer supplemented with protease inhibitor cocktail (Roche), and then immunoprecipitated with FLAG or HA affinity agarose (Sigma-Aldrich) overnight at 4 °C. Beads containing affinity-bound proteins were washed six times with immunoprecipitation wash buffer (150 mM NaCl, 10 mM HEPES pH 7.4, 0.1% NP-40), followed by elution with 1 M glycine (pH 3.0) twice. The eluted proteins were denatured and separated on SDS-polyacrylamide gels and stained with Coomassie blue; the indicated bands were subjected to MS analysis. Purification of recombinant proteins His-RFC4 or HA-NICD1 was performed using immunoprecipitation as described previously.

**Notch signal activity analysis.** The Notch signaling activity was measured by dual-luciferase reporter assay (Promega) of Notch1 signaling activity reporter, which is constructed by cloning the DNA-binding motif of NICD1 (CBF1/RBP-Jκ binding site: 4 × CCGTGGGAAAAAATTT) into pGL3-Basic plasmid as a promoter of Firefly Luciferase gene, and Renilla Luciferase reporter (TK plasmid) used as an internal control. Relative luciferase activity (Firefly Luciferase/Renilla Luciferase) of each treatment is calculated as the Notch signaling activity.

**SPR kinetic analysis.** SPR kinetic analysis was performed with a method suggested by the instruction manual using HBS-P running buffer (10 mM HEPES, pH 7.4, 150 mM NaCl, 0.05% (v/v) surfactant P20) at 25 °C and employing a series S sensor chip CM5 on the BiaCore T200 SPR instrument (GE Healthcare). The recombinant FLAG-tagged RFC4 protein was diluted in the running buffer and then immobilized to a density of 600–770 response units. The recombinant HA-tagged NICD1 protein (at concentrations 30–500 nM) was injected on the chip surface for 180 s at a flow rate of 20 µl/min, with the dissociation phase monitored for up to 600 s. Individual sensor grams were double-referenced against injection onto an empty flow cell and HA-alone injections at equivalent concentrations. Data were fitted to a 1:1 Langmuir model using the BIAevaluate 4.0.1 analysis software.

**Chromatin immunoprecipitation (ChIP).** For ChIP analysis, $5 \times 10^{7}$ 293FT or A549 cells cultured in 15-cm culture dishes were harvested for cross-linking and sheared by sonication. The resultant chromatin fraction was immunoprecipitated using 10 µg antibodies against H3K27ac or RBP-Jκ (Abcam) or negative control anti-IgG (Sigma-Aldrich). After reversing the cross-links with NaCl and removing proteins with proteinase K, enriched DNA fragments were purified and isolated via phenol/chloroform extraction and ethanol precipitation. The final DNA pellets were then subjected to real-time quantitative PCR with the indicated specific primers.

**Pulse-chase analysis.** Transfect target gene plasmid into NSCLC cells. After 36 h, wash the cells with pre-warmed phosphate-buffered saline (PBS) (37 °C) to remove residual unlabeled amino acids. Replace PBS by pre-warmed MEM, supplemented with arginine, leucine, glucose, inositol, 0.2% BSA, and 10 mM HEPES (pH 7.4). Cultivate cells for 15 min at 37 °C, 5% $CO_2$ in a water-saturated atmosphere. Add 4.3 MBq TRAN $^{35}$S-LABEL (MP Biomedicals, Santa Ana, CA, USA) to the medium and cultivate cells for 30 min at 37 °C, 5% $CO_2$ in a water-saturated atmosphere. Wash the cells with pre-warmed PBS and change medium to standard medium supplemented with 5% fetal bovine serum (FBS).

Then harvest cells after various periods of time (chase). Aspirate the medium and wash the cells two times in ice-cold PBS, Subsequent to harvesting cells at the final time point place the frozen cell culture dishes on ice and cover the cells with 500 µl RIPA lysis buffer. Then immunoprecipitated your interested protein with 1 µg of specific antibody and separated proteins on a standard discontinuous SDS gel. Incubate the gel in fixation buffer (40% methanol, 10% acetic acid, 50% distilled water) for 30 min at room temperature. Wash the gel for 30 min in $H_2O$, then incubate the gel in 1 M sodium salicylate for 30 min. Put the gel on 3MM paper and cover the gel with Saran wrap. Then dry the gel at 70 °C in a vacuum dryer until it is completely free of water. Finally, detect the radioactive protein by phosphorimager (Storm 840, GE Healthcare).

**In vitro ubiquitination assay.** Cells were transfected with various combinations of plasmids or siRNAs as indicated, along with HA-tagged ubiquitin (Ub). At 24 h after transfection, the cells were treated with MG132 (10 µM) for 6 h, and the whole-cell lysates prepared with NP-40-containing lysis buffer were subjected to

immunoprecipitation for exogenous FLAG-tagged NICD1 proteins. The levels of NICD1 ubiquitination were detected by immunoblotting with anti-FLAG antibody.

**Cell invasion assay.** Indicated cells ($4 \times 10^4$) were plated on the top side of Transwell chambers (Corning) pre-coated with matrigel (BD Biosciences) and incubated at 37 °C for 24–36 h, followed by removal of cells inside the upper chamber with cotton swabs. Cells invading to the bottom side of the membrane were fixed, stained, photographed and quantified in in five random ×200 magnification fields.

**Tumor sphere culture.** Indicated cells ($2.5 \times 10^3$), seeded in ultra-low adherent six-well plates (Corning), were cultured in DMEM/F12 serum-free medium (Invitrogen) supplemented with 2% of B27, 20 ng/ml of EGF (BD Biosciences), 20 ng/ml of bFGF, and 4 mg/ml insulin (Sigma-Aldrich) to form tumor spheres. Nutrient supplemented medium was added for the growth of spheres every 2 days for 10 days. Cell spheres were photographed and counted under ×200 magnification.

**Flow cytometry analysis.** For side-population (SP) analysis, indicated cells were dissociated with trypsin and re-suspended at $1 \times 10^6$ cells/ml in DMEM containing 2% FBS and then incubated at 37 °C for 30 min with or without 50 μM Verapamil (Sigma-Aldrich) to inhibit ABC transporters and to confirm the side-population cells. The cells were subsequently incubated with 5 μg/ml Hoechst33342 (Sigma-Aldrich) for 90 min at 37 °C, plated on ice for 10 min, washed with ice-cold PBS and subjected to flow cytometry analysis using flow cytometer BD influx (BD Biosciences) as instructed by the manufacturer. For cell cycle analysis, synchronization of cells was facilitated with serum starvation overnight. Cells were fixed in EtOH (70%) overnight at 4 °C followed by extraction of DNA in DNA-Extraction buffer (0.2 M Na$_2$HPO$_4$, pH 7.8; 0.1% Triton) for 5 min at RT, and staining of DNA in staining-buffer (20 μg/ml PI + 200 μg RNaseA) for 15 min at 37 °C. Cell cycle phase was checked by flow cytometry analysis using flow cytometer BD influx. And cell apoptosis was evaluated with Annexin V Apoptosis Detection Kit APC (eBioscience).

**Animal models.** Female BALB/c-nu mice (5–6 weeks of age, 18–20 g) and female C57BL/6N (5–6 weeks of age, 18–20 g) mice were housed in specific pathogen-free facilities on a 12 h light/dark cycle at temperature 18–22 °C and humidity 50–60%. To investigate the effects of RFC4 and Notch signaling activation on tumor distant metastasis or lung colonization, the indicated luciferase-expressing cells ($0.5–1 \times 10^6$) were intracardially or intravenously injected, and metastases were monitored by bioluminescent imaging every 3 days. For bioluminescent imaging assay, 15 min prior to imaging, mice were injected intraperitoneally (i.p.) with 150 mg/kg luciferin. Following general anesthesia, images were taken and analyzed with Spectrum Living Image 4.2 software (Caliper Life Sciences). To assess the self-renewal abilities of NSCLC cells in the tumorigenicity model, indicated cells of various dosages ($5 \times 10^5$, $5 \times 10^4$ and $5 \times 10^3$) were subcutaneously inoculated. Tumor lengths (L) and widths (W) were measured every week using a digital caliper and tumor volumes (V) were calculated using the formula $V = L \times W^2/2$. At the indicated experimental endpoints, mice were anesthetized and sacrificed, and tumors as well as various organs (brain, lung, liver, and bones) were resected, sectioned (5 μm in thickness), and histologically examined by H&E staining. H&E images were captured using the AxioVision Rel. 4.6 computerized image analysis system (Carl Zeiss). All animal studies were approved by the SYSU Institutional Animal Care and Use Committee.

**Immunohistochemistry (IHC) assay.** IHC assays using anti-NICD1, anti-RFC4, or anti-HES1 were separately conducted on paraffin-embedded specimens of NSCLC patients. The immunostaining intensities of indicated proteins was evaluated and scored by two independent observers, scoring both the proportions of positive staining tumor cells and the staining intensities. Scores representing the proportion of positively stained tumor cells was graded as 0 (no positive tumor cells), 1 (<10%), 2 (10–50%), and 3 (>50%). The staining intensity was determined as 0 (no staining); 1 (weak staining = light yellow), 2 (moderate staining = yellow brown), and 3 (strong staining = brown). The staining index (SI) was calculated as staining intensity × percentage of positive tumor cells, resulting in scores as 0, 1, 2, 3, 4, 6, and 9. Cut-off values for high- and low-expression of protein of interest were chosen based on a measurement of heterogeneity using the log-rank test with respect to overall survival.

**Fluorescence in situ hybridization (FISH).** Dual-color FISH assays were conducted using following probes: RFC4/CEP11 probe mixture containing homebrewed RFC4 DNA labeled with SpectrumOrange (Vysis, Inc.) and the chromosome 11 control probe CEP11 (centromere enumeration) labeled with SpectrumGreen (Vysis, Inc.). Whole-tissue sections were deparaffinized, boiled, digested, and incubated with the FISH probe and slides were then sealed with rubber cement[58]. Following a denaturation step, slides were incubated overnight at 37 °C, followed by wash five times in 2× SSC buffer containing 0.3% NP-40 and counterstaining with 1 μg/ml 4,6-diamidino-2-phenylindole (DAPI) (Sigma-Aldrich). FISH RFC4 gene amplification was defined as both RFC4/CEP11 ratio ≥2

and RFC4 gene copy number ≥4. Both criteria were required to be met to rule out samples with RFC4/CEP11 ratio ≥2 merely due to isolation loss of CEP11.

**Statistical analysis.** The correlation between RFC4 expression and clinicopathologic characteristics was analyzed by the chi-square test. The survival curve was established by the Kaplan–Meier method and compared by the log-rank test. All statistical analyses except the sequencing data were performed using the PASW Statistics 18 version 18.0.0 (SPSS Inc., Chicago, IL, USA) software package and GraphPad Prism 8 version 8.3.0 (GraphPad Software, San Diego, CA, USA). Comparisons between two groups were performed using Student's t-test (two-tailed), while analyses comparing multiple treatments with a control group were performed using two-way ANOVA with post hoc Dunnett's multiple comparisons test. All error bars represent the mean ± SD derived from three independent experiments. In all cases, $P < 0.05$ was considered to be statistically significant.

**Study approval.** All experimental procedures and use of NSCLC donors' samples were approved by the SYSU Institutional Animal Care and Use Committee, and Research Ethics Committee. Donors provided prior written informed consent.

**Reporting summary.** Further information on research design is available in the Nature Research Reporting Summary linked to this article.

## Data availability
The TCGA Lung Adenocarcinoma (LUAD) and Lung Squamous Cell Carcinoma (LUSC) sequencing data used in this study are available in a public repository from the GDC Data Portal Data Release Version 20.0 [https://portal.gdc.cancer.gov/]. The RNA-sequencing data that support the findings of this study has been deposited in GEO with the accession code GSE137106. All other data supporting the findings of this study are available from the corresponding author upon reasonable request. Source data are provided with this paper.

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

## Acknowledgements

This work was supported by the National Key Research and Development Program of China (2017YFA0106300); the Key Program of the National Natural Science Foundation of China (81820108025); The Foundation for Innovative Research Groups of the National Natural Science Foundation of China (81621004); the Natural Science Foundation of China (81330058, 81772473, 81472351, 81902965, 81802274); the Guangdong Basic and Applied Basic Research Foundation (2016A030306026, 2014A030306023, 2018A0303130260, 2019A1515011174); the Science and Technology Program of Guangzhou City (201803010039, 201804010057); the Fundamental Research Funds for the Central Universities (19ykpy162); the Chongqing Youth Talent Support Program (CQYC2020057957); the Youth Talent Support Program of the First Affiliated Hospital of Chongqing Medical University (BJRC2020-01).

## Author contributions

L.L., T.T., and S. Liu participated in the design of the study and carried out experiments. X.Y., X. Chen, J. Liang, R.H., W.W., Y.Y., X.L., Y.Z., Q.L., S. Liang, H.Y., Y.W., X.G., Y.L., X.D., H.G., J.W., X.Z., J.Y., J. Li, S.S., and M.L. conceived the experiments, analyzed the data, and provided advices and help on the concept and experiments of this project. H.T., J.C., and X. Cai conceived of the study, and participated in its design and coordination and critical review of the manuscript. All authors were involved in writing the paper and had final approval of the submitted and published versions.

## Competing interests

The authors declare no competing interests.
