## [Peer Review File · Nature Communications]

Reviewers' comments:

Reviewer #2 (Remarks to the Author):

The authors provide evidence to support a novel role for RFC4 in blocking CDK8-mediated degradation of NICD, and that this results in enhanced invasiveness/stem-ness in NSCLC cells in vitro/vivo. The mechanistic link between RFC4 over-expression (common in NSCLC) and increased NICD-dependent gene expression is strongly suggested to be a result of direct binding of RFC4 to NICD. A significant piece of data absent from the manuscript is the impact of genetic manipulation e.g RFC4 over-expression on cell growth rates/cell cycle distribution (see comments below). The experimental data on the impact of high RFC4 expression have strong correlates with prognosis in NSCLC clinical datasets. A limitation of the current data is that expression of RFC4 strongly correlates with expression of other DNA replication and proliferation associated genes (reviewer's own analysis of TCGA datasets used in this study), and therefore a causative effect of RFC4 over-expression on clinical outcomes cannot be definitively made, despite the strong correlations. Nonetheless, this is the first report that this reviewer is aware of of this unexpected role of RFC4, and it potentially has impact across other disease sites where RFC4 is over-expressed. The manuscript would benefit from careful scientific proofreading. There are a number of specific comments below.

Fig 1a

Not clear which samples are 'LN metastasis' (dichotomised?). Not clear what the red/blue graded scale represents if LN mets are present or absent? How were samples ranked from left-to-right?

Fig 1b

Scale bar not defined (throughout manuscript)

Fig 1c

RNA or protein expression? What was cut-off to define high/low? Appears to be some kind of median.

What is the effect of NICD1 over-expression on in vitro growth rate/cell cycle/apoptosis?

Fig 1i

What is scale showing? Where is the list of significantly altered genes?

Fig 1j

What level of RFC4 knockdown was achieved with shRNA? Did this affect cell growth rate in vitro? Did silencing any of the other genes upregulated by NICD1 have any effect? Or was only RFC4 tested?

Fig 1l

The xenograft growth curves show a remarkable lack of variability in size between animals within each group (also for Fig 5h) especially given the small group sizes (n=5). It would be reassuring to see individual animal growth curves. Need to state what error bars represent (SD, SEM?) throughout results.

The table (and other similar tables) needs more explanation in the legend. What are 4/5, 1/5 etc?

Fig 1 Legend

States that error bars represent +/-SD from three independent experiments, but this surely does not apply to the in vivo experiments

Results Page 6

Authors use the term 'silenced' for RFC4 expression, but do not indicate what level of knockdown (RFC4 protein) that this corresponds to.

Results Page 8

It is not clear how the authors define RFC4 upregulation in 103/105 cases. What is the cut-off?

Why is the cut-off there?

The effect of RFC4 over-expression or knockdown on cell proliferation, cell cycle and apoptosis in

cancer cell lines would be informative to determine their contribution to the phenotypes reported.

Fig 4e

More details of the Notch signalling activity assay needed. Include level of knockdown achieved with RFC4 siRNA

Results Page 9

The authors state that the expression levels of RFC4 positively correlated with levels of several canonical downstream genes of Notch signalling (Fig 4j). However, the authors do not report which of the ~20,000 genes most correlate with RFC4 expression. In particular, are genes associated with DNA synthesis, cell cycle etc better correlated with RFC4 than canonical Notch signalling

Results Page 10

It's not clear whether increased expression of RFC4 is associated with increased expression of CDK8 in clinical samples. If they are co-upregulated, then the hypothesis that increased RFC4 expression leads to increased NICD1 levels becomes weaker

Results Page 11

RFC4 is an essential gene for growth/survival of many cell lines (<https://depmap.org/>) and therefore proliferation within the sh-RFC4 xenograft (Fig 6d) may have selected for cells with only marginally reduced RFC4 levels which would affect how these data are interpreted.

Figure 6 g/h

It is not clear whether NICD1 is wild-type version or stabilised version

Figure 7b

It is not standard to display data for just top and bottom 25% of cases. Data for all quartiles should be shown

Figure 7d-f, h

Needs to be more precise in figure or legend which datasets are used for each figure, and the rationale for defining high/low (7d-f) (median?)

Reviewer #3 (Remarks to the Author):

The manuscript titled "A RFC4/Notch1 signaling feedback loop promotes NSCLC metastasis and stemness" by Liu et al., demonstrated a RFC4 mediated regulation of Notch1 activity which is a novel finding has a high clinical significance. This study identified RFC4 as a novel regulator of Notch1 and showed a pathological role associated with NSCLC metastasis and stemness. In silico analysis of data obtained from tumor samples has identified an upregulation of RFC4 and NICD1 in metastatic tumors that are significantly correlated with patient overall survival. Overall, this work is novel and highly interesting however addressing the below mentioned concerns definitely will increase the quality of the work.

Major weaknesses of this study include:

1. Using only a single lung tumor cell line as well as 293 cells to demonstrate the majority of these effects. Experiments should include a panel of lines with different expression of these proteins.
2. Using transfected highly overexpressed NICD to study the role of RFC4 mediated Notch1 regulation and associated pathological functions. It would be difficult to envision a role for RFC4 in Notch1 regulation in significant patient population without a demonstration under physiological conditions in multiple cancer cell lines.
3. In this study, using patient data, the authors identified a strong correlation between RFC4 expression to metastasis which needs to be validated in a panel of cell lines with low vs high endogenous RFC4 expression and xenograft experiments. Cell lines with high RFC4 expression should correlate in vivo metastasis and Notch1 regulation compared to low RFC4 expression in a

physiological states. Alternatively, a demonstration of endogenous RFC4 knockdown causes a decrease in metastasis and stemness in multiple high-expressing NSCLC cell lines would be important.

4. 3q amplification is mostly seen in squamous cancers where Notch is thought to be a tumor suppressor. This should be addressed in the discussion.

Minor concerns:

Methods are not adequately reported to really understand some of the key findings presented in this paper.

The statement about the 1% 5 year survival of patients with metastatic disease on page 3 is likely outdated in the era of immunotherapy in which we are seeing 15 – 20% 5-year survivals.

Figure. 1b: How was NICD1 levels were determined in IHC staining? Scoring of IHC was not clear. Figure. 1b: NICD1 low and NICD1 high levels where these total Notch1 levels or NICD1 levels. How was NICD1 levels were determined?

Figure 2a: bottom panel requires NICD1 levels to show that JAG1 treatment induces Notch activation.

Figure 4d: show RFC4 western analysis demonstrating the knockdown.

Figure e, g, h and i: please indicate what is NC

Figure 7a: Correlation of RFC4 expression and Notch1 expression should be determined to establish the claimed connection between NICD1 and RFC4.

The grammar needs editing.

The figures have too many panels.

Reviewer #4 (Remarks to the Author):

Aberrant Notch signaling has long been established to play a key role in tumor formation. Notch is normally activated and cleaved at the membrane to release the intra-cellular domain (ICD) transcription factor. The most prominent oncogenic downstream target of Notch1 is probably c-Myc, a driver of nearly every tumor. But also critical inhibitors of differentiation are activated by Notch. Hyper-activation of Notch1 usually occurs via point mutations that either constitutively activate or stabilize the protein. The ICD is normally degraded by the Fbxw7 ubiquitin ligase via a degron in the C-terminus of Notch. In cancers, this regulatory mechanism is often eliminated by frame-shift mutations that truncate the protein prematurely, especially in T-cell leukemia. Therefore, inhibition of Notch signaling has received great attention in patient treatment. However, while gamma-secretase inhibitors work well in cell culture, they are intolerable in clinical settings, and alternative approaches are needed.

RFC4 is one of several subunits of the replication factor C complex and functions in DNA replication and repair as a polymerase accessory protein. It has been found over-expressed in many cancer studies and is associated with tumor progression and poor prognosis.

In this manuscript, the authors identify RFC4 as Notch1 activated gene that cooperates with the Notch ICD in tumor metastasis. The RFC4 gene is amplified in a large percentage of non-small cell lung cancers and directly activated by Notch1. Mechanistically, RFC4 is shown here to directly bind and stabilize the Notch1 ICD by preventing its phosphorylation, presumably at the degron site, and thereby evade recognition by the Fbxw7 ubiquitin ligase.

This is an impressive study, and the data presented are very interesting and important to the field.

Indeed, a quick look at available cancer data in c-Bioportal for NSCLC reveals near mutually exclusive RFC4 amplification, activating Notch1 mutations, and inactivating Fbw7 mutations, suggesting they function complementary on the same pathway. But while some data nicely establish the authors's model, other data do not sufficiently support their claims and need additional work, especially Figures 4 & 5. Moreover, it is difficult to reconcile how a clamp loader (RFC4) for a DNA polymerase would have such stabilizing effect on the NICD. While not impossible, this discrepancy should be discussed better.

Other comments:

1. Several stemness genes were analyzed for RFC4 responsiveness upon NICD expression (Fig. 1J). Surprisingly, c-Myc is not presented here (nor L-Myc). Why not? How about other genes not implicated in stemness? I would also like to see a control in the set that is not inhibited by RFC4 knockdown to ensure that such knockdown doesn't interfere with (NICD-dependent) transcription in general. As this figure appears, virtually every gene they looked at is regulated by RFC4.
2. In Fig. 4a it is shown that RFC4 specifically induces the NICD, but not full-length Notch1, suggesting a post-translational mechanism like protein stabilization of the NICD. The full-length band of Notch is rather poorly shown. How do the authors know this band is indeed Notch1? Please, provide a knock-down to establish this is the correct band.
3. The IP in Fig 4c should be performed in reverse (IP anti HA, blot with Flag!) as the HA blot could represent any Notch-associated ubiquitinated protein.
4. The CHX chase looks rather unimpressive in Fig 4d. CHX can interfere with the assay itself, and it should first be established that it doesn't by using a standard pulse chase assay.
5. The IPs in Fig. 5b and 5c are completely uncontrolled. Even though IgG lanes are widely used as controls, I much prefer bait knock-downs, and the authors should use knock-downs instead in 5b and leave out the bait in 5c to demonstrate specificity of their interactions. They could also co-express both proteins in cells and do the IPs without using recombinant protein, along with the according controls.
6. Fig 5f is insufficient to make the point of competitive binding between CDK8 and RFC4 to NICD. No cyclin was used in this assay, which is known to greatly alter the conformation of the kinase. And how about the other way around? Do increasing amounts of CDK8 displace RFC4? Best is to examine the NICD - RFC4 interaction as in 5b and knock-down CDK8, or examine the NICD - CDK8 interaction with RFC4 knock-down.
7. The NICD is full of potential serine phosphorylation sites. By using a general p-Ser but otherwise not site-specific antibody, the authors have no idea what they are looking at. Sure, the site seems CDK8 responsive (Fig 5h & i). But the relevant site in the Notch degron is actually a threonine (T2512). The contributing effects of S2514 and S2517 are controversial at best. Without knowing the site in question, there is no evidence for a mechanism.
8. The point of Fig 5k is to demonstrate that enforced RFC4 expression displaces CDK8 from Notch complexes, which then would abolish degron phosphorylation and therefore diminish Fbxw7 binding. We have extensive experience with Fbxw7 in my lab. I therefore find these data unbelievable. For one, Fbxw7 protein is notoriously hard to detect by itself, let alone in co-IPs. But more importantly: without uncoupling of binding from turnover, a ubiquitin ligase, like Fbw7, cannot be co-immunoprecipitated by a substrate as it would become degraded upon contact. Now, there are some exceptions. But Notch isn't one of them. In addition, it is our experience that the antibody used in this study (3A9) makes an Fbxw7-independent cross-reaction right underneath where untagged Fbw7 would migrate on a gel. Therefore, this experiment needs to be better controlled using Fbxw7 knock-downs, or better crispr-mediated knockout. There are also cell lines available with targeted deletion of Fbw7.
9. For Fig 5l, see my comments in 3. You could literally look at Fbxw7 auto-ubiquitylation, but sell it for Notch.

Overall, this is good work. The functional link of NICD and RFC4 is well illuminated. But the mechanism is still rather unclear.

Reviewer #2 (Remarks to the Author):

The authors provide evidence to support a novel role for RFC4 in blocking CDK8-mediated degradation of NICD, and that this results in enhanced invasiveness/stem-ness in NSCLC cells in vitro/vivo. The mechanistic link between RFC4 over-expression (common in NSCLC) and increased NICD-dependent gene expression is strongly suggested to be a result of direct binding of RFC4 to NICD. A significant piece of data absent from the manuscript is the impact of genetic manipulation e.g RFC4 over-expression on cell growth rates/cell cycle distribution (see comments below). The experimental data on the impact of high RFC4 expression have strong correlates with prognosis in NSCLC clinical datasets. A limitation of the current data is that expression of RFC4 strongly correlates with expression of other DNA replication and proliferation associated genes (reviewer's own analysis of TCGA datasets used in this study), and therefore a causative effect of RFC4 over-expression on clinical outcomes cannot be definitively made, despite the strong correlations. Nonetheless, this is the first report that this reviewer is aware of this unexpected role of RFC4, and it potentially has impact across other disease sites where RFC4 is over-expressed. The manuscript would benefit from careful scientific proofreading. There are a number of specific comments below.

1. Fig 1a

Not clear which samples are 'LN metastasis' (dichotomised?). Not clear what the red/blue graded scale represents if LN mets are present or absent? How were samples ranked from left-to-right?

Response: We thank the reviewer for raising these issues. The Gene Set Enrichment Analysis (GSEA) was employed to analyze whether the priori defined set of genes (Notch signaling up-regulated genes) shows statistically significant, concordant differences between different lymph node metastasis statuses of patients in TCGA NSCLC cohort. These NSCLC samples were divided into two subgroups, namely, with lymph node metastasis (LN metastasis) subgroup or without lymph node metastasis (non-LN-metastasis) subgroup, which were derived from 302 and 407 patients, respectively, according to their lymph node metastasis status. In the GSEA analysis, the genes were ranked according to the fold-changes of their expression between the two subgroups from high to low (the graded scale from left-to-right). The red scale represents genes highly expressed in the LN metastasis subgroup and the blue scale represents genes highly

expressed in the non-LN metastasis subgroup. Each black straight line represents a gene in priori defined gene set and the position of line represents its ranking in the ordered gene list. In addition, an enrichment score (ES) of each gene in the ordered gene list was calculated to create a running ES score. The running ES score increases when the current gene is present in the Notch signaling up gene set and highly-expressed, otherwise it decreases. The final ES score was the maximum value of the running ES score (i.e., 0.49 in Figure 1a). Therefore, the data analyzed in Figure 1a showed that the majority of Notch signaling up-regulated genes were enriched and upregulated in the LN metastasis subgroup as compared to the non-LN-metastasis subgroup, suggesting that Notch signaling is activated in NSCLC samples with lymph node metastasis. In this revision, we have re-analyzed the updated RNA-seq profiles of the TCGA NSCLC cohort using the GSEA assay and found significant correlation between Notch signaling activation and LN metastasis. Corresponding data have been incorporated into the revised manuscript on Page 6.

2. Fig 1b

Scale bar not defined (throughout manuscript)

Response: The scale bars have been incorporated into the revised figures/supplementary figures and defined in the figure/supplementary figure legends throughout the manuscript in this revision.

3. Fig 1c

RNA or protein expression? What was cut-off to define high/low? Appears to be some kind of median.

Response: We thank the reviewer for pointing out the unclear description for Figure 1c. The Kaplan-Meier analysis is based on the protein levels of NICD1 assessed by IHC assays in our own collected 219 cases of NSCLC specimens. The immunostaining intensities of NICD1 was evaluated and scored by two independent observers, scoring both the proportions of positive staining tumor cells and the staining intensities. Scores representing the proportion of positively stained tumor cells was graded as: 0 (no positive tumor cells), 1 (<10%), 2 (10-50%), and 3 (>50%). The staining intensity was determined as: 0 (no staining); 1 (weak staining = light yellow), 2 (moderate staining = yellow brown), and 3 (strong staining = brown). The staining index (SI) was calculated as (staining

intensity) × (percentage of positive tumor cells), resulting in scores as 0, 1, 2, 3, 4, 6 and 9. Cut-off value for high- and low-expression of NICD1 was chosen based on the median value of SI. The detailed information has been incorporated into the revised Supplementary Methods.

4. What is the effect of NICD1 over-expression on in vitro growth rate/cell cycle/apoptosis?

Response: The authors thank the reviewer for raising this important issue. We performed a series of in vitro assays to evaluate the effect of NICD1 overexpression in NSCLC cells on cell growth, cell cycle and apoptosis. As shown in Supplementary Figure 1 a-e for Reviewer 2, NICD1 overexpression significantly increased cell growth rate, promoted cell colony formation, anchor-independent growth and cell cycle progression and inhibited cisplatin-induced apoptosis, which is consistent with the important roles of overactivated Notch1 signaling in tumor cell proliferation and survival. As expected, silencing RFC4, which is a DNA replication factor, not only markedly reversed the effects of NICD1 overexpression on promoting invasion/migration and cancer stemness, but also reversed NICD1-enhanced tumor cell proliferation and survival (new Supplementary Figure 1, a-g). Moreover, silencing RFC4 significantly compromised the invasive and self-renewal abilities of NICD1-overexpressing NSCLC cells even when cell proliferation was inhibited by mitomycin C (a DNA synthesis inhibitor) treatment or cell death was induced by cisplatin treatment, respectively, evaluated in the cell invasion assay and tumor sphere culture assay (new Supplementary Figure 1, h and i). These data suggest that RFC4 plays crucial roles in promoting overactivated Notch1-driven tumor metastasis and stemness, which can be independent of the promoting effects of NICD1 on tumor cell proliferation and survival. Also notably, we also proved that the potent pro-metastatic and pro-self-renewal effects of RFC4 can be independent of its role in DNA replication and repair and its ability to enhance cell proliferation/cell cycle/survival as described below (new Supplementary Figures 4 and 5). Moreover, primary NSCLC tumors exhibiting LN metastasis expressed much higher levels of NICD1 or RFC4 than non-metastatic tumors (new Figure 1b and Figure 7b). Therefore, our study focuses on the important regulatory roles of a RFC4/Notch1 signaling feedback loop in NSCLC metastasis and stemness. Corresponding data have been incorporated into the revised manuscript on Pages 6, 10 and 15.

5. Fig 1i

What is scale showing? Where is the list of significantly altered genes?

Response: The graded red/blue scale for the heatmap in Figure 1i shows the range of log₁₀ FPKM (Fragments Per Kilobase of transcript per Million mapped reads) values, which is a normalized estimation of gene expression based on the RNA-sequencing data, for each gene significantly deregulated in A549-Vector and A549-NICD1 cells (fold change \geq 2, P <0.05). The FPKM values of RFC4 in A549-NICD1 and A549-Vector cells are 286700.5 and 2675.689, respectively. To present the heatmap data more clearly, in this revision, top 14 upregulated genes in A549-NICD1 cells as compared to A549-Vector cells are provided in the heatmap (new Figure 1i). Furthermore, the list of significantly altered genes between A549-NICD1 and A549-Vector cells is provided as new Supplementary Data 1 in this revision. Appropriate description has been incorporated into the revised manuscript on Pages 6 and 21.

6. Fig 1j

What level of RFC4 knockdown was achieved with shRNA? Did this affect cell growth rate in vitro?

Did silencing any of the other genes upregulated by NICD1 have any effect? Or was only RFC4 tested?

Response: The authors thank the reviewer for raising these important questions. As measured by western blotting and calculated by gray scan of immunoblots, shRNA against RFC4 in RFC4-silenced A549-NICD1 cells efficiently decreased protein levels of RFC4 to less than 30%, of those in A549-NICD1 and A549-Vector cells (new Supplementary Figure 1, a and j). As expected, silencing RFC4, which is a DNA replication factor, not only markedly reversed the effects of NICD1 overexpression on promoting invasion/migration and cancer stemness, but also reversed NICD1-enhanced tumor cell growth rate, proliferation and survival (new Supplementary Figure 1, a-g). Moreover, silencing RFC4 significantly compromised the invasive and self-renewal abilities of NICD1-overexpressing NSCLC cells even when cell proliferation was inhibited by mitomycin C (a DNA synthesis inhibitor) treatment or cell death was induced by cisplatin treatment (new Supplementary Figure 1, h and i). These data suggest that RFC4 plays crucial roles in promoting

overactivated Notch1-driven tumor metastasis and stemness, which can be independent of the promoting effects of RFC4 on tumor cell proliferation and survival. Corresponding data have been incorporated into the revised manuscript on Pages 6 and 7.

To address the reviewer's concern, we had evaluated the biological importance of the top 10 NICD1 up-regulated genes, namely, RFC4, BOLA2B, PROK1, SLUG, KCNA5, HES5, TFF1, ANKRD1, SPRY4 and CALCB in NICD1-induced malignancy. As shown in Supplementary Figure 1 f-l for Reviewer 2, while silencing RFC4 or SLUG expression markedly reversed the promoting effects of NICD1 overexpression on cell growth, invasion and stemness of NSCLC cells, knockdown of BOLA2B, PROK1, KCNA5, ANKRD1 or CALCB reversed NICD1-induced NSCLC cell growth at various degrees, but hardly compromised NICD1-induced cell invasion or stemness, and HES5 knockdown apparently inhibited self-renewal, mildly inhibited cell growth, and hardly inhibited cell invasion of NICD1-overexpressing NSCLC cells. Due to too many figures/supplementary figures in this revision, we prefer to not present the abovementioned data in the revised manuscript, and we will present them if the reviewer feels necessary.

7. Fig 11

The xenograft growth curves show a remarkable lack of variability in size between animals within each group (also for Fig 1h) especially given the small group sizes (n=5). It would be reassuring to see individual animal growth curves. Need to state what error bars represent (SD, SEM?) throughout results.

The table (and other similar tables) needs more explanation in the legend. What are 4/5, 1/5 etc?

Response: The authors thank the reviewer for raising the suggestion. In this revision, the individual tumor growth curve of each mouse subcutaneously inoculated with indicated cells has been provided (new Figures 1h, 1l and 3e, Supplementary Figure 3l, 5i, 8d and 8g). Error bars in black represent mean \pm SD of volumes of visible/detectable tumors within each group (nude mice without formation of subcutaneous tumors are excluded from the calculation of mean and SD). The tables show the subcutaneous tumor formation frequency of 5 mice inoculated with indicated cells and cell numbers. For example, in Figure 1l, when separately inoculated with 5×10^5 , 5×10^4 and 5×10^3 cell numbers, A549-NICD1-Scramble cells formed visible/detectable tumors in 5, 4, and 3 out of 5 mice, respectively, contrasting that 4, 1 and 0 out of 5 mice inoculated with 5×10^5 , 5×10^4 and

5×10^3 cell numbers of A549-Vector-Scramble cells developed subcutaneous tumors, and silencing RFC4 greatly abrogated the subcutaneous tumor formation frequency of NICD1-overexpressing NSCLC cells. Appropriate description has been incorporated into the corresponding figure legends in the revised manuscript.

8. Fig 1 Legend

States that error bars represent +/-SD from three independent experiments, but this surely does not apply to the in vivo experiments.

Response: We are sorry for the unclear description in the figure legends. The error bars only in Figure 1j represent mean \pm SD derived from three independent experiments, whereas the error bars in Figures 1g, 1h, 1k and 1l represent mean \pm SD derived from technical replicates in each group. We have re-checked the figure legends to avoid misunderstanding and modified the corresponding figure legends in the revised manuscript.

9. Results Page 6

Authors use the term 'silenced' for RFC4 expression, but do not indicate what level of knockdown (RFC4 protein) that this corresponds to.

Response: To address the reviewer's concern, western blotting analysis showed that protein levels of RFC4 in A549-luci-NICD1 cells silenced for RFC4 expression were decreased to about 17% and 43%, respectively, of those in A549-luci-NICD1-Scramble and A549-luci-Scramble. Similarly, RFC4 protein levels in subcutaneous tumors formed by RFC4-silenced A549-luci-NICD1 cells were decreased to 16% and 47%, respectively, of those in subcutaneous tumors formed by A549-luci-NICD1-Scramble and A549-luci-Scramble cells (new Supplementary Figure 1, j and k). The grey scan values of RFC4 immunoblot bands relative to those of GAPDH immunoblot bands have been provided to indicate the level of RFC4 knockdown in this revised manuscript.

10. Results Page 8

It is not clear how the authors define RFC4 upregulation in 103/105 cases. What is the cut-off? Why is the cut-off there?

Response: We thank the reviewer for pointing out the unclearness here. Originally, fold

change of Tumor / Normal >1 was considered as RFC4 upregulation and thus 103 out of 105 cases expressed increased RFC4 mRNA levels in NSCLC tissue as compared to paired normal lung tissue. In this revision, however, fold change of Tumor / Normal >2 (log₂ fold change >1, a general threshold value) is set as the new cut-off value, therefore 81 out of 105 cases can be defined as RFC4 upregulation in NSCLC tissue as compared to paired normal tissue. We have added a dash line at log₂ fold change (Tumor / Normal) = 1 representing the new cut-off value in the new Supplementary Figure 3a. Corresponding modification has been made the revised manuscript.

11. The effect of RF4C over-expression or knockdown on cell proliferation, cell cycle and apoptosis in cancer cell lines would be informative to determine their contribution to the phenotypes reported.

Response: The reviewer's suggestion is well taken. In this revision, MTT, colony formation, soft agar, FACS of cell cycle, and Annexin V assays were performed to evaluate the biological effects of RFC4 on cell proliferation, cell cycle and apoptosis. As shown in new Supplementary Figure 4, a-e, overexpression of RFC4 significantly promoted, whereas knockdown of RFC4 markedly suppressed NSCLC cell proliferation/growth, cell cycle progression and resistance to cisplatin-induced cell apoptosis. Moreover, overexpressing RFC4 significantly potentiated, whereas silencing RFC4 impaired the invasive and self-renewal abilities of NSCLC cells even when cell proliferation was inhibited by mitomycin C treatment or cell death was induced by cisplatin treatment, respectively, evaluated in the cell invasion assay and tumor sphere culture assay (new Supplementary Figure 4, f and g). Furthermore, since RFC4 is one of several subunits of the replication factor C complex, consisting of RFC1, RFC2, RFC3, RFC4 and RFC5, which function as a clamp loader for recruiting PCNA to DNA and assisting in DNA replication and repair as polymerase accessory proteins, we also evaluated the impact of inhibition of DNA replication on RFC4-induced tumor invasion and stemness. As shown in new Supplementary Figure 5, b-i, silencing the essential DNA replication accessory gene PCNA significantly reversed the promoting effect of RFC4 on cell proliferation, but failed to interfere RFC4-induced Notch1 signaling activation, or RFC4-potentiated tumor invasion and stemness in vitro or tumor metastasis and tumorigenicity in vivo; by contrast, silencing Notch1 greatly reversed all the tested phenotypes potentiated by RFC4 in NSCLC cells both in vitro and in vivo. Notably, the TCGA lung cancer datasets show that the mRNA levels of RFC2 and RFC5,

which bind RFC4 to form a core polymerase accessory complex, are averagely upregulated about 1.6-1.8 folds in LUAD and 2.6-3.5 folds in LUSC, as compared to adjacent normal tissues, and the mRNA levels of RFC1 are slightly down-regulated in the TCGA LUAD and LUSC datasets (Supplemental Figure 1m for Reviewer 2). By contrast, the mRNA levels of RFC4 are averagely increased about 2.93 folds in LUAD and 9.42 folds in LUSC in the same TCGA lung cancer datasets (new Supplementary Figure 9e). Indeed, unlike RFC4, protein levels of RFC2 or RFC5 were rarely upregulated in NSCLC tissue as compared to adjacent normal lung tissue (new Supplementary Figure 5a). Moreover, silencing RFC2 or RFC5 reversed the promoting effects of RFC4 on NSCLC cell proliferation and cell cycle progression, but failed to interfere RFC4-induced Notch1 signaling activation, or RFC4-potentiated tumor invasion and stemness in vitro or tumor metastasis and tumorigenicity in vivo (new Supplementary Figure 5, b-i). Taken together, these results strongly suggest that RFC4 potently promotes tumor invasion/metastasis and stemness/tumorigenicity, which can be independent of its role in DNA replication and repair and its ability to enhance cell proliferation/cell cycle/survival. Corresponding data have been incorporated into the revised manuscript on Page 10.

12. Fig 4e

More details of the Notch signaling activity assay needed. Include level of knockdown achieved with RF4C siRNA

Response: In this revision, more details of the Notch signaling activity are provided in the Supplemental Methods. Briefly, the Notch signaling activity was measured by dual-luciferase activity assay of Notch1 signaling reporter, which is constructed by cloning the DNA binding motif of NICD1 (CBF1/RBP-J κ binding site: 4 \times CCGTGGGAAAAATTT) into pGL3-Basic plasmid as a promoter of Firefly Luciferase gene, and Renilla luciferase reporter (TK plasmid) used as an internal control. Relative luciferase activity (Firefly Luciferase / Renilla Luciferase) of each treatment is calculated as the Notch signaling activity. The efficiency of distinct shRNAs against RFC4 in the indicated NSCLC cells ranged from 70% to 95%, as evaluated by Western Blotting and grey scan of both RFC4 and GAPDH immunoblot bands. The grey scan values of RFC4 immunoblot bands relative to those of GAPDH immunoblot bands have been provided to indicate the level of RFC4 knockdown in this revised manuscript. Appropriate information of the Notch signaling activity assay has been incorporated into the revised manuscript.

13. Results Page 9

The authors state that the expression levels of RFC4 positively correlated with levels of several canonical downstream genes of Notch signaling (Fig 4j). However, the authors do not report which of the ~20,000 genes most correlate with RFC4 expression. In particular, are genes associated with DNA synthesis, cell cycle etc better correlated with RFC4 than canonical Notch signaling.

Response: The authors thank the reviewer for raising this important comment. Indeed, genes most correlated with RFC4 expression are associated with DNA synthesis and cell cycle, such as *CCNB2* ($r = 0.80$; $P < 0.001$), *PCNA* ($r = 0.75$; $P < 0.001$), *CDK2* ($r = 0.75$; $P < 0.001$) and *POLR2H* ($r = 0.75$; $P < 0.001$), as analyzed from the TCGA lung cancer datasets, which consist of 321 stage I cases, 223 stage II cases, 144 stage III cases and 21 stage IV cases (Supplementary Figure 2a for Reviewer 2). We guess that the majority of NSCLC tissues with early-stage diseases might contribute to the most correlation between RFC4 and genes associated with DNA synthesis and cell cycle. Interestingly, when these 709 cases of NSCLC samples are divided into Non-LN metastasis ($n = 407$) and LN metastasis subgroups ($n = 302$), in the LN metastasis subgroup, the correlations of RFC4 expression with canonical downstream genes of Notch signaling or metastasis/stemness related genes, such as *NRARP* ($r = 0.65$; $P < 0.001$), *HEY1* ($r = 0.54$; $P < 0.001$), *MYC* ($r = 0.48$; $P < 0.001$), *SOX2* ($r = 0.66$; $P < 0.001$), *SLUG* ($r = 0.55$; $P < 0.001$), are close to those with genes associated with DNA synthesis and cell cycle, such as *CCNB2* ($r = 0.73$; $P < 0.001$), *PCNA* ($r = 0.68$; $P < 0.001$), *CDK2* ($r = 0.69$; $P < 0.001$), *POLR2H* ($r = 0.70$; $P < 0.001$) and *MKI67* ($r = 0.41$; $P < 0.001$). Notably, the number of metastatic NSCLC samples is quite small relative to the number of non-metastatic NSCLC samples, which might largely affect the analysis that whether genes associated with DNA synthesis and cell cycle are better correlated with RFC4 than canonical Notch signaling. Due to too many data in this revision, we prefer to not present the correlation data in the revised manuscript, and we will present them if the reviewer feels necessary.

14. Results Page 10

It's not clear whether increased expression of RFC4 is associated with increased expression of CDK8 in clinical samples. If they are co-upregulated, then the hypothesis that increased RFC4 expression leads to increased NICD1 levels becomes weaker.

Response: The authors thank the reviewer for raising this interesting point. From the TCGA lung cancer datasets, the mRNA levels of both CDK8 and CCNC (CDK8 binding cyclin) are slightly increased in NSCLC tissue as compared to normal lung tissue whereas CDK8 protein levels are rarely upregulated in NSCLC tissue; by contrast, both mRNA and protein levels of RFC4 are significantly upregulated in NSCLC tissue as compared to normal lung tissue (Supplementary Figure 2b for Reviewer 2; new Supplementary Figure 5a). Thus, RFC4 and CDK8 are not co-upregulated in NSCLC tissue and increased expression of RFC4 is not associated with the expression of CDK8 in clinical NSCLC samples. Importantly, the SPR kinetic analysis showed that the binding affinity between RFC4 and NICD1 was approximately five-folds higher than that between CDK8 and NICD1 (new Figure 5c). Moreover, the amounts of CDK8 or its binding partner Cyclin C, were gradually impaired and even totally diminished in the pulled down proteins when HA-tagged NICD1 proteins were immunoprecipitated following addition of purified RFC4 in a dose-dependent manner; interestingly, although the binding of NICD1 to RFC4 could be impaired by the increasing amounts of purified CDK8 in a dose-dependent manner, it appears that large amounts of CDK8 proteins are required to significantly abrogate the interaction between NICD1 and RFC4 (new Figure 5d). Furthermore, while silencing RFC4 mildly enhanced the interaction between NICD1 and CDK8, silencing CDK8 significantly enhanced the interaction between NICD1 and RFC4; in parallel, overexpressing RFC4 significantly abrogated the binding of NICD1 to CDK8 whereas CDK8 overexpression mildly impaired the binding of NICD1 to RFC4 (new Figure 5e). These data strongly suggest that both higher binding affinity of RFC4 to NICD1 and increased expression levels of RFC4 in NSCLC tissue make RFC4 more competitive than CDK8 to bind to NICD1. Therefore, in NSCLC increased expression of RFC4 able to directly bind NICD1 should competitively abrogate CDK8/FBXW7-mediated degradation of NICD1 and thus lead to increased NICD1 levels. Corresponding data have been incorporated into the revised manuscript on 12 and 13.

15. Results Page 11

RFC4 is an essential gene for growth/survival of many cell lines (<https://depmap.org/>) and therefore proliferation within the sh-RFC4 xenograft (Fig 6d) may have selected for cells with only marginally reduced RFC4 levels which would affect how these data are interpreted.

Response: We thank the reviewer for this interesting comment. To address the reviewer's concern, in this revision, we evaluated the levels of RFC4 and NICD1 expression and cell proliferation/apoptosis in subcutaneous tumors formed by A549-Vector-Scramble, A549-NICD1mut-Scramble, or A549-NICD1mut-sh-RFC4 cells. As shown in new Figure 6d, subcutaneous tumors formed by A549 cells overexpressing stabilized mutant of NICD1, which is resistant to RFC4 silencing-caused CDK8/FBXW7-dependent degradation of NICD1, presented significantly increased levels of RFC4, CCND1 and PCNA and decreased proportion of TUNEL-positive tumor cells, as compared to those formed by control A549 cells; whereas subcutaneous tumors formed by RFC4-silenced A549-NICD1mut cells presented remarkable reduction in RFC4 protein levels and only marginally altered levels of NICD1, CCND1 or PCNA expression or apoptotic tumor cell proportion. These molecular changes in subcutaneous tumors are consistent with the observed phenotypic changes, namely, silencing RFC4 could not compromise the potent pro-metastatic and pro-tumorigenic abilities of the NICD1-mutant NSCLC cells, but significantly reversed the aggressiveness of NSCLC cells transduced with wild-type NICD1. The *in vitro* and *in vivo* data further support our hypothesis that the RFC4-induced stabilization of NICD1 plays a pivotal role in promoting NSCLC metastasis and stemness properties. Therefore, we guess that the potent tumor-promoting effects of stabilized mutant of NICD1 can totally compensate the effects of silencing RFC4 on NSCLC cell growth and survival. Corresponding data have been incorporated into the revised manuscript on Pages 14 and 15.

16. Figure 6 g/h

It is not clear whether NICD1 is wild-type version or stabilized version.

Response: In new Figure 6 f and g, which were presented as Figure 6 g and h in the initial manuscript, NICD1 is wild-type version. In contrast to the remarkable suppressive effects of DAPT (γ -secretase inhibitor) treatment in Notch1-overexpressing NSCLC cells on their invasion and stemness properties because DAPT blocks the generation of NICD1 from γ -secretase-dependent cleavage of Notch1, DAPT treatment hardly reversed NICD1- or RFC4-induced invasion and stemness, probably because high NICD1 and RFC4 levels form a positive feedback loop (NICD1 induces RFC4 transcription and RFC4 stabilizes NICD1 proteins), which should no longer require γ -secretase to cleave Notch1 into NICD1 and thus be insensitive to DAPT treatment. Besides, all the cell lines used in our study

harbor wild-type Notch1. Appropriate description has been incorporated into the figure legends in the revised manuscript.

17. Figure 7b

It is not standard to display data for just top and bottom 25% of cases. Data for all quartiles should be shown.

Response: Per the reviewer's suggestion, the expression levels of stemness- and metastasis-associated genes, namely, ALDH1A1, ALDH3A1, SOX2, CD44, CK18, CDH2, SLUG and TWIST1 in 971 cases of primary NSCLC tumors from the TCGA lung cancer datasets stratified according to all four quartiles of RFC4 expression are provided as new Supplementary Figure 9a, showing that NSCLC tissues with higher RFC4 expression have significantly higher levels of these stemness- and metastasis-associated genes than those with lower RFC4 expression. Similarly, the expression data for several Notch signaling downstream genes, including HES1, HEY1, HEY2 and NRARP, in new Supplementary Figure 6i (presented as Supplemental Figure 3c in the initial version of the manuscript) are also provided according to all four quartiles of RFC4 expression in the 971 cases of primary NSCLC tumors from the TCGA lung cancer datasets. Corresponding data and description have been incorporated into the revised manuscript.

18. Figure 7d-f, h

Needs to be more precise in figure or legend which datasets are used for each figure, and the rationale for defining high/low (7d-f) (median?)

Response: We thank the reviewer for the suggestion. The datasets used for original Figure 7, d and e, which are provided as new Figure 7c and Supplementary Figure 9b in this revision, are from our own collected cohort of 219 cases of NSCLC patients, for original Figure 7f, which is provided as new Supplementary Figure 9c in this revision, are from the online kmpplot database (<http://kmpplot.com/>), for original Figure 7g, which is provided as new Supplementary Figure 9d in this revision, are from the MSKCC lung cancer datasets downloaded from Ladanyi and Gerald Laboratories (http://cbio.mskcc.org/public/lung_array_data/), and for original Figure 7h, which is provided as new Figure 7d in this revision, are from the TCGA lung cancer datasets. The rationale for defining high/low of RFC4 or NICD1 to evaluate its correlation with overall

survival, progression-free survival or LN metastasis-free survival, is based on corresponding median values of RFC4 protein or mRNA levels or NICD1 protein levels. Appropriate description has been incorporated into the figure legends in the revised manuscript.

Supplementary Figure 1 for Reviewer 2

Supplementary Figure 2 for Reviewer 2

a

Table. Pearson correlation coefficient of RFC4 with cell cycle and Notch-related genes' expression in TCGA NSCLC Dataset

Genes	Total		Non-LN metastasis		LN metastasis	
	Pearson's R	P value	Pearson's R	P value	Pearson's R	P value
PCNA	0.75	<0.001	0.81	<0.001	0.68	<0.001
MKI67	0.52	<0.001	0.59	<0.001	0.41	<0.001
CDK2	0.75	<0.001	0.79	<0.001	0.69	<0.001
POLR2H	0.75	<0.001	0.77	<0.001	0.70	<0.001
CCNB2	0.80	<0.001	0.83	<0.001	0.73	<0.001
HES1	0.24	<0.001	0.15	<0.001	0.29	0.007
HEY1	0.47	<0.001	0.39	<0.001	0.54	<0.001
NRARP	0.59	<0.001	0.51	<0.001	0.65	<0.001
SOX2	0.60	<0.001	0.53	<0.001	0.66	<0.001
ALDH1A1	0.31	<0.001	0.22	<0.001	0.42	<0.001
MYC	0.42	<0.001	0.35	<0.001	0.48	<0.001
SLUG	0.50	<0.001	0.46	<0.001	0.55	<0.001

b

Reviewer #3 (Remarks to the Author):

The manuscript titled “A RFC4/Notch1 signaling feedback loop promotes NSCLC metastasis and stemness” by Liu et al., demonstrated a RFC4 mediated regulation of Notch1 activity which is a novel finding has a high clinical significance. This study identified RFC4 as a novel regulator of Notch1 and showed a pathological role associated with NSCLC metastasis and stemness. In silico analysis of data obtained from tumor samples has identified an upregulation of RFC4 and NICD1 in metastatic tumors that are significantly correlated with patient overall survival. Overall, this work is novel and highly interesting however addressing the below mentioned concerns definitely will increase the quality of the work.

Major weaknesses of this study include:

1. Using only a single lung tumor cell line as well as 293 cells to demonstrate the majority of these effects. Experiments should include a panel of lines with different expression of these proteins.

Response: The authors greatly appreciate the reviewer’s encouraging comment and take the reviewer’s suggestion seriously. In this revision, we employed two highly metastatic lung cancer cell lines, H1975 and LLC (Lewis lung cancer cells), and cultured primary lung cancer cells (LC1) originated from a stage III LAD patient’s primary lung tumor, all of which express high levels of NICD1 and RFC4, to further demonstrate the pivotal roles of the positive feedback loop between RFC4 and NICD1 in coupling NSCLC metastasis and stemness properties. As expected, activating the Notch signaling by overexpressing NICD1 or by JAG1 treatment significantly increased RFC4 expression in low-metastatic NSCLC cell line A549, whereas inhibiting the Notch1 activation by silencing Notch1 or by treatment with DAPT, a γ -secretase inhibitor, decreased RFC4 expression at both protein and mRNA levels in multiple highly metastatic NSCLC cells, namely, H1975, LLC and LC1 (new Figure 2, a and b; Supplementary Figure 2, a and b). Moreover, activating or inhibiting the Notch1 signaling, respectively, promoted and abrogated an enrichment of activated histone H3K27ac in the promoter region of *RFC4* gene in these low- or high-metastatic NSCLC cells (new Figure 2, c and d; Supplementary Figure 2c). In parallel, silencing RBP-J κ also significantly reduced transcription and thus expression of RFC4 in H1975, LC1 and LLC cells (new Figure 2, e and f; Supplementary Figure 2, d and e). Moreover, ChIP and luciferase reporter assays revealed a binding of RBP-J κ to the

predicted site in the upstream promoter region of *RFC4* gene in various NSCLC cells (new Figure 2, h and i; Supplementary Figure 2, f and g). Additionally, using H1975, LC1 and LLC cells with endogenous high levels of RFC4 and NICD1, we further demonstrate that RFC4 could tightly bind with NICD1 and competitively inhibits CDK8-mediated phosphorylation and FBXW7-mediated polyubiquitination of NICD1, resulting in stabilization of NICD1 proteins, especially in the nucleus, as evidenced by the dramatic effects of RFC4 silencing on promoting CDK8/FBXW7-dependent NICD1 degradation and reducing NICD1 levels (new Figures 4, 5 and 6; Supplementary Figures 6 and 7). These data strongly support our conclusion that RFC4 is a new target gene of the Notch signaling and that high levels of RFC4 stabilize NICD1 to form a positive feedback and are enough to cause overactivation of the Notch1 signaling, probably without requiring γ -secretase-dependent cleavage of Notch1 into NICD1, in NSCLC progression. Corresponding data have been incorporated into the revised manuscript. Of note, due to too many panels in figures, we have removed the majority of data derived from 293FT cells in this revision and we would like to present them if the reviewer feels necessary.

2. Using transfected highly overexpressed NICD to study the role of RFC4 mediated Notch1 regulation and associated pathological functions. It would be difficult to envision a role for RFC4 in Notch1 regulation in significant patient population without a demonstration under physiological conditions in multiple cancer cell lines.

Response: The authors thank the reviewer for raising this important point. In this revision, we employed highly metastatic lung cancer cell lines, H1975 and LLC, and metastatic lung cancer cells (LC1) primarily cultured as mentioned above, all of which express high levels of RFC4 and NICD1, to further evaluate the role for RFC4 in Notch1 regulation. As shown in new Figure 4a-e, and Supplementary Figure 6a-f, silencing RFC4 in these metastatic lung cancer cells significantly decreased NICD1 protein levels in the nucleus without affecting Notch1 mRNA levels or the quantities of full-length Notch1 proteins or cytoplasmic NICD1 proteins, increased K48-linked polyubiquitination of NICD1, caused shortened half-lives of NICD1 proteins and thus inhibited transcriptional activity of the Notch signaling. Moreover, RFC4 could tightly bind with NICD1 and abrogated the interaction between CDK8 or FBXW7 with NICD1; therefore, silencing RFC4 significantly increased CDK8-dependent serine and threonine phosphorylation of nuclear NICD1 proteins and failed to cause K48-linked polyubiquitination of NICD1 when

FBXW7 or CDK8 was pre-silenced or depleted in H1975, LC1 and LLC cells (new Figure 5, a-k; Supplementary Figure 7, a-f). Notably, per Reviewer 4's request, we also prove that both higher binding affinity of RFC4 to NICD1 and increased expression levels of RFC4 in NSCLC tissue make RFC4 more competitive than CDK8 to bind to NICD1 using various experimental models (new Figure 5, c-e; Supplementary Figure 5a). Additionally, also per Reviewer 4's request, our data suggest that the binding of RFC4 to the PEST domain of NICD1 might squeeze away other NICD1-interactive proteins, including various NICD1 kinases, resulting in abrogation of NICD1 phosphorylation at multiple serine or threonine amino acids, such as S2514, S2517, T2512, and T2542, in the PEST domain (new Supplementary Figure 7, g and h). Taken together, using human and murine lung cancer cell lines, as well as primary lung cancer cells from a clinic patient, we strongly demonstrate an important role for RFC4 in promoting NICD1 stabilization and over-activating the Notch1 signaling through competitively inhibiting Notch1 phosphorylation and FBXW7-mediated polyubiquitination of NICD1 in NSCLC. Corresponding data have been incorporated into the revised manuscript.

3. In this study, using patient data, the authors identified a strong correlation between RFC4 expression to metastasis which needs to be validated in a panel of cell lines with low vs high endogenous RFC4 expression and xenograft experiments. Cell lines with high RFC4 expression should correlate *in vivo* metastasis and Notch1 regulation compared to low RFC4 expression in a physiological state. Alternatively, a demonstration of endogenous RFC4 knockdown causes a decrease in metastasis and stemness in multiple high-expressing NSCLC cell lines would be important.

Response: The authors take the reviewer's suggestion seriously. In this revision, we employed highly metastatic lung cancer cell lines, H1975 and LLC, and metastatic primary lung cancer cells LC1 as mentioned above to evaluate the effect of silencing RFC4 on tumor metastasis and stemness both *in vitro* and *in vivo*. As shown in new Figure 3, a and b, and Supplementary Figure 3, e-i, in contrast to the pro-invasive and pro-self-renewal effects of overexpressing RFC4 in low-metastatic NSCLC cells A549 and H1703, which express moderate levels of RFC4, silencing RFC4 dramatically compromised the invasive and self-renewal abilities of highly metastatic lung cancer cells namely, H1975, LLC and LC1, which express high levels of RFC4. In *in vivo* studies, in contrast to the pro-metastatic and pro-tumorigenic effects of overexpressing RFC4 in low-metastatic NSCLC

cells A549, RFC4-silenced H1975 cells presented much weakened metastatic bioluminescent signals and developed much fewer and smaller metastatic lesions in the lung or brain of nude mice, as well as displayed greatly compromised tumorigenicity ability, as compared to vector-control H1975 cells when they were separately injected intracardially or intravenously (new Figure 3, c and d; Supplementary Figure 3, j-l). Moreover, we used immunocompetent mice (C57BL/6) to establish experimental metastasis and tumorigenicity models of RFC4-silenced and vector-control LLC cells. Our results showed that silencing RFC4 significantly suppressed the ability of highly metastatic murine lung cancer cells to form lung metastases when injected intravenously or to develop subcutaneous tumors when injected with various cell numbers ranging from 5×10^3 to 5×10^5 in C57BL/6 mice (new Figure 3, f and g). Notably, we recently have established bone-metastasis models of lung cancer cells PC9 and H460, obtained cell derivatives PC9BM and H460BM specifically prone to bone metastasis and found that protein levels of RFC4, but not NICD1, are markedly up-regulated in bone-metastasis cells (Supplementary Figure 3, a and b, for Reviewer 3), indicating other distinct mechanisms possibly underlying RFC4-induced bone metastasis. This is an ongoing project and we prefer to not present more details here. Taken together, our data strongly demonstrate the essential role of RFC4 in conferring lung cancer cells both metastasis and stemness properties. Corresponding data have been incorporated into the revised manuscript on Page 9.

4. 3q amplification is mostly seen in squamous cancers where Notch is thought to be a tumor suppressor. This should be addressed in the discussion.

Response: The authors greatly thank the reviewer for raising this important issue and the suggestion. In many types of squamous cancers, such as squamous cell carcinoma of the head and neck, skin, oral cavity, and esophagus, 10%-20% tumor cases harbor inactivating mutations in the Notch1 gene and many studies suggest the tumor-suppressive roles of Notch1 in these squamous cancers^{1, 2, 3}. In lung squamous cell carcinoma (LUSC), approximately 5% of patient samples harbor inactivating mutations in the Notch1 gene⁴. However, both the oncogenic and tumor-suppressive roles of Notch1 have been reported in LUSC^{5, 6}. Interestingly, as analyzed from the TCGA lung cancer datasets and KM plot database, high mRNA levels of Notch1 significantly correlate with poor overall survival and disease progression of LUSC patients (Supplementary Figure 3, c-e, for Reviewer 3),

indicating that Notch1 may play important tumor-promoting roles during LUSC progression. On the other hand, in consistent with the notion that amplification of 3q chromosome, where the RFC4 gene is located, is mostly seen in squamous cancers, the mRNA levels of RFC4 are averagely increased about 9.42 folds in LUSC, approximately 40.3% of which have RFC4 gene amplification in the TCGA lung cancer datasets; by contrast, RFC4 mRNA levels are averagely increased about 2 folds in LUAD and approximately 2.33% of LUAD samples have RFC4 gene amplification in the TCGA lung cancer datasets. Our study demonstrates that Notch1 signaling activation induces direct transcription of RFC4 and RFC4 proteins bind to stabilize NICD1 proteins, forming a positive feedback loop between high RFC4 and NICD1 levels and sustained overactivation of Notch signaling, which leads to lung cancer tumorigenicity and metastasis. These data suggest that both the transcriptional upregulation of RFC4 by activated Notch1 signaling and RFC4 amplification should contribute to high levels of RFC4 to varying degrees in LUAD and LUSC, resulting in lung cancer progression. Also notably, the binding of RFC4 to NICD1 might alter the preference of NICD1 binding to the promoters/enhancers of downstream target genes of the Notch1 signaling, as silencing RFC4 reversed NICD1-induced expression of its downstream genes at various degrees (Figure 1j; Supplementary Figure 6h). Therefore, we conclude that both LUAD and LUSC could utilize the pivotal roles of the positive feedback loop between RFC4 and NICD1 in coupling NSCLC metastasis and stemness properties. Appropriate information has been incorporated into the Discussion Section in the revised manuscript on Page 7, 11, 19 and 20.

Minor concerns:

5. Methods are not adequately reported to really understand some of the key findings presented in this paper.

Response: We thank the reviewer for pointing out the issue. More detailed description of methods has been added into the Materials and Methods section in the revised manuscript. Of note, as limited by manuscript length, some detailed information of materials and methods are shown in Supplementary Materials.

6. The statement about the 1% 5 years survival of patients with metastatic disease on page 3 is likely outdated in the era of immunotherapy in which we are seeing 15~20% 5-year survivals.

Response: We are sorry for the careless error and thank the reviewer for pointing out it. Corrections have been made in the revised manuscript on Page 3.

7. Figure.1b: How was NICD1 levels were determined in IHC staining? Scoring of IHC was not clear.

Figure.1b: NICD1 low and NICD1 high levels where these total Notch1 levels or NICD1 levels. How was NICD1 levels were determined?

Response: We thank the reviewer for raising these important questions. In our IHC staining assays, the antibody (ab8925, abcam) only detects the activated Notch1 fragment (aa 1755-1767) as the corresponding epitope, which is exposed after γ -secretase cleavage of Notch1 into NICD1, and thus only detects NICD1 but is unable to recognize the un-cleaved full length Notch1. The immunostaining intensities of NICD1 proteins were evaluated and scored by two independent observers, scoring both the proportions of positive staining tumor cells and the staining intensities. Scores representing the proportion of positively stained tumor cells were graded as: 0 (no positive tumor cells), 1 (<10%), 2 (10-50%), and 3 (>50%). The staining intensity was determined as: 0 (no staining); 1 (weak staining = light yellow), 2 (moderate staining = yellow brown), and 3 (strong staining = brown). The staining index (SI) was calculated as staining intensity \times percentage of positive tumor cells, resulting in scores as 0, 1, 2, 3, 4, 6 and 9. Cutoff value for high-expression (> median value) and low-expression (\leq median value) of NICD1 was chosen based on the median value of SI, which is 4 in 219 cases of NSCLC specimens collected in this study. Corresponding information has been provided in the revised manuscript.

8. Figure 2a: bottom panel requires NICD1 levels to show that JAG1 treatment induces Notch activation.

Response: Per the reviewer's advice, in this revision, NICD1 expression levels in new Figure 2a are provided showing that JAG1 treatment induces γ -secretase-dependent cleavage of Notch1 and thus generation of NICD1, which can be abrogated by addition of the γ -secretase inhibitor DAPT. As a downstream target gene of the Notch1 signaling, RFC4 expression is also significantly upregulated by JAG1 treatment and inhibited by addition of DAPT. Corresponding data have been incorporated into the revised manuscript.

9. Figure 4e: show RFC4 western analysis demonstrating the knockdown.

Figure 4e, g, h and i: please indicate what is NC

Response: The efficiency of RFC4 knockdown in various NSCLC cells evaluated by western analysis has been provided in new Figure 4a and Supplementary Figure 6f for new Figure 4e and Supplementary Figure 6e. In original Figure 4e, g, h and i in the initial version of manuscript, NC represents for ‘negative control’, which means scramble siRNAs. In this revision, we re-performed these experiments to evaluate the effects of silencing RFC4 on Notch1 signaling activity in various RFC4 stably silenced NSCLC cells (per the Reviewer’s request as above mentioned) and consistent found that silencing RFC4 inhibited transcriptional activity of the Notch signaling (new Figure 4e) and greatly reversed the activation of the Notch signaling pre-induced by NICD1 or JAG1 in NSCLC cells, and the Notch signaling in NSCLC cells pre-silenced with RFC4 became insensitive to stimulation of JAG1 overexpressed in HUVEC cells (new Supplementary Figure 6, e and g). Corresponding modification has been incorporated into the revised manuscript on Page 11.

10. Figure 7a: Correlation of RFC4 expression and Notch1 expression should be determined to establish the claimed connection between NICD1 and RFC4.

Response: The authors thank the reviewer for raising this suggestion. Using the TCGA lung cancer datasets, we analyzed the correlation of RFC4 expression and both Notch1 expression and the Notch1 signaling. As shown in Supplementary Figure 2b for Reviewer 2, RFC4 mRNA levels are not significantly correlated with Notch1 mRNA levels (the Notch1 signaling activation may be more relied on the generation and accumulation of NICD1 proteins). However, high RFC4 mRNA levels positively correlated with enriched transcription of downstream genes of the Notch1 signaling as evidenced by the GSEA analysis (new Figure 4f), suggesting the positive correlation between RFC4 mRNA levels and the Notch1 signaling. In particular, we had shown that expression levels of RFC4 positively correlated with the levels of several canonical downstream genes of the Notch1 signaling, including HES1, HEY1, HEY2 and NRARP, and patients within the lower quartile of RFC4 expression expressed lower levels of each of these downstream genes in their lung tumors than those within the higher quartile of RFC4 expression (new Figure 4g; Supplementary Figure 6i). In consistence, we confirmed the positive correlation between RFC4 protein levels and protein levels of NICD1 and HES1 in 219 cases of NSCLC

specimens collected in this study (new Figure 4h). These data suggest the tight connection between RFC4 expression with NICD1 levels and the Notch1 signaling activation. Corresponding data have been incorporated into the revised manuscript on Page 11.

11. The grammar needs editing. The figures have too many panels.

Response: The authors thank the reviewer for this very helpful advice. This revised manuscript has been 'spell-checked' and 'grammar-checked' carefully by editing service prior to re-submission. The revised figures have been modified to avoid too many panels by moving many of them into the supplementary figures, as well as removing the majority of data derived from 293FT cells in this revision.

References

1. Stransky N, *et al.* The Mutational Landscape of Head and Neck Squamous Cell Carcinoma. *Science* **333**, 1157-1160 (2011).
2. Agrawal N, *et al.* Comparative Genomic Analysis of Esophageal Adenocarcinoma and Squamous Cell Carcinoma. *Cancer Discovery* **2**, 899-905 (2012).
3. Egloff AM, Grandis JR. Molecular Pathways: Context-Dependent Approaches to Notch Targeting as Cancer Therapy. *Clin Cancer Res* **18**, 5188-5195 (2012).
4. Westhoff B, *et al.* Alterations of the Notch pathway in lung cancer. *P Natl Acad Sci USA* **106**, 22293-22298 (2009).
5. Zou B, Zhou XL, Lai SQ, Liu JC. Notch signaling and non-small cell lung cancer (Review). *Oncol Lett* **15**, 3415-3421 (2018).
6. Wang NJ, *et al.* Loss-of-function mutations in Notch receptors in cutaneous and lung squamous cell carcinoma. *P Natl Acad Sci USA* **108**, 17761-17766 (2011).

Supplementary Figure 3 for Reviewer 3

Reviewer #4 (Remarks to the Author):

Aberrant Notch signaling has long been established to play a key role in tumor formation. Notch is normally activated and cleaved at the membrane to release the intra-cellular domain (ICD) transcription factor. The most prominent oncogenic downstream target of Notch1 is probably c-Myc, a driver of nearly every tumor. But also critical inhibitors of differentiation are activated by Notch. Hyper-activation of Notch1 usually occurs via point mutations that either constitutively activate or stabilize the protein. The ICD is normally degraded by the Fbxw7 ubiquitin ligase via a degron in the C-terminus of Notch. In cancers, this regulatory mechanism is often eliminated by frame-shift mutations that truncate the protein prematurely, especially in T-cell leukemia. Therefore, inhibition of Notch signaling has received great attention in patient treatment. However, while gamma-secretase inhibitors work well in cell culture, they are intolerable in clinical settings, and alternative approaches are needed.

RFC4 is one of several subunits of the replication factor C complex and functions in DNA replication and repair as a polymerase accessory protein. It has been found over-expressed in many cancer studies and is associated with tumor progression and poor prognosis.

In this manuscript, the authors identify RFC4 as Notch1 activated gene that cooperates with the Notch ICD in tumor metastasis. The RFC4 gene is amplified in a large percentage of non-small cell lung cancers and directly activated by Notch1. Mechanistically, RFC4 is shown here to directly bind and stabilize the Notch1 ICD by preventing its phosphorylation, presumably at the degron site, and thereby evade recognition by the Fbxw7 ubiquitin ligase.

This is an impressive study, and the data presented are very interesting and important to the field. Indeed, a quick look at available cancer data in c-Bioportal for NSCLC reveals near mutually exclusive RFC4 amplification, activating Notch1 mutations, and inactivating Fbxw7 mutations, suggesting they function complementary on the same pathway. But while some data nicely establish the authors' model, other data do not sufficiently support their claims and need additional work, especially Figures 4 & 5. Moreover, it is difficult to reconcile how a clamp loader (RFC4) for a DNA polymerase would have such stabilizing effect on the NICD. While not impossible, this discrepancy should be discussed better.

Response: The authors greatly appreciate the reviewer's encouraging comments and constructive suggestions. In this revision, per reviewers' request, we provide much more

data to strengthen our claims, especially for Figures 4&5 as addressed below, including usage of lung cancer cells primarily cultured from NSCLC patients to clarify the feedback loop between high RFC4 and NICD1 levels and sustained overactivation of Notch signaling, and the importance of RFC4-induced NICD1 stability in promoting NSCLC tumorigenicity and metastasis under physiological conditions. Corresponding data have been incorporated into the revised manuscript.

Notably, RFC4 is one of several subunits of the replication factor C (RFC) complex, consisting of RFC1, RFC2, RFC3, RFC4 and RFC5, which function in DNA replication and repair as polymerase accessory proteins. Notably, the TCGA lung cancer datasets show that the mRNA levels of RFC2 and RFC5, which bind RFC4 to form a core polymerase accessory complex, are averagely upregulated about 1.6-1.8 folds in LUAD and 2.6-3.5 folds in LUSC, as compared to adjacent normal tissues, and the mRNA levels of RFC1 are slightly down-regulated in the TCGA LUAD and LUSC datasets. By contrast, the mRNA levels of RFC4 are averagely increased about 2.93 folds in LUAD and 9.42 folds in LUSC in the same TCGA lung cancer datasets. In consistence, protein levels of RFC2 and RFC5 are rarely upregulated in NSCLC tissue as compared to adjacent normal lung tissue whereas RFC4 are significantly up-regulated in NSCLC (new Supplementary Figure 5a), Moreover, silencing RFC2 or RFC5 reversed the promoting effects of RFC4 on NSCLC cell proliferation and cell cycle progression, but neither interfered RFC4-enhanced NICD1 levels and Notch1 signaling activation, nor impaired RFC4-potentiated tumor invasion and stemness in vitro or tumor metastasis and tumorigenicity in vivo (new Supplementary Figure 5, b-i), suggesting that the effects of RFC4 on stabilizing NICD1 and promoting tumor invasion/metastasis and stemness/tumorigenicity can be independent of its role in DNA replication and repair and its ability to enhance cell proliferation/cell cycle/survival. We guess that minimal levels of RFC4, as well as other subunits of the RFC complex, can maintain enough DNA synthesis for unlimited growth of tumor cells, and redundant levels of RFC4 may have distinct molecular and biological functions. Corresponding data and appropriate information have been incorporated into the revised manuscript on Page 10.

Other comments:

1. Several stemness genes were analyzed for RFC4 responsiveness upon NICD expression (Fig. 1J). Surprisingly, c-Myc is not presented here (nor L-Myc). Why not? How about other genes not implicated in stemness? I would also like to see a control in the set that is

not inhibited by RFC4 knockdown to ensure that such knockdown doesn't interfere with (NICD-dependent) transcription in general. As this figure appears, virtually every gene they looked at is regulated by RFC4.

Response: We thank the reviewer for raising these interesting points. Although overexpressing NICD1 significantly upregulated c-Myc expression, but not L-Myc, silencing RFC4 mildly reversed NICD1-induced c-Myc expression (new Figure 1j). In this revision, we also evaluated the effect of silencing RFC4 on expression of a set of downstream genes of Notch1 signaling according to previous reports, such as VEGF, LUNAR, IL7R, ID1, DTX1, HES1, HES4, HES5, HEY1, HEY2, NRARP, CCND1, BCL-2 and BCL-XL, in NICD1-overexpressing NSCLC cells. As shown in new Supplementary Figure 6g, silencing RFC4 significantly reversed NICD1-induced expression of VEGF, LUNAR, IL7R, ID1, HES1, HES5, HEY1, HEY2, NRARP, CCND1, BCL-2 and BCL-XL at various degrees. However, mRNA levels of HES4 and DTX1 were significantly upregulated by NICD1, but were not reversed by RFC4 silencing in NICD1-overexpressing NSCLC cells, indicating that RFC4 knockdown doesn't globally interfere with NICD-dependent transcription (new Supplementary Figure 6h). Appropriate data have been incorporated into the revised manuscript on Page 11.

2. In Fig. 4a it is shown that RFC4 specifically induces the NICD, but not full-length Notch1, suggesting a post-translational mechanism like protein stabilization of the NICD. The full-length band of Notch is rather poorly shown. How do the authors know this band is indeed Notch1? Please, provide a knock-down to establish this is the correct band.

Response: Per the reviewer's request, in this revision, we evaluated the effects of silencing RFC4 or Notch1 on protein levels of Notch1 and NICD1. As shown in new Figure 4a, as the knockdown of Notch1 validated the correct band of Notch1 and could be used as a positive control, silencing RFC4 significantly decreased, whereas RFC4 overexpression greatly enhanced NICD1 protein levels without affecting the quantities of full-length Notch1 proteins in multiple highly metastatic NSCLC cells, including human and murine lung cancer cell lines H1975 and LLC, and cultured primary lung cancer cells (LC1) originated from a stage III LAD patient's primary lung tumor, per Reviewer 3's request. These data indeed suggest a post-translational mechanism like protein stabilization of the NICD. We have re-provided the protein levels of full-length Notch1, which are not affected by RFC4 overexpression or knockdown in various NSCLC cells, whereas

silencing Notch1 with two distinct shRNAs in H1975 cells strikingly diminishes protein levels of full-length Notch1 (new Figure 4a). Corresponding modification have been incorporated into the revised manuscript.

3. The IP in Fig 4c should be performed in reverse (IP anti HA, blot with Flag!) as the HA blot could represent any Notch-associated ubiquitinated protein.

Response: The authors thank the reviewer for this useful suggestion. The IP assays immunoprecipitated with anti-HA affinity agarose showed that the Flag-tagged NICD1 (NICD1-Flag) was only pulled down by HA-tagged wildtype K48-linked ubiquitin (UbK48-HA), but not by HA-tagged K48 mutated ubiquitin (UbK48R-HA) (new Supplementary Figure 6b). Moreover, knockdown of RFC4 enhanced the interaction between UbK48-HA and NICD1-Flag, further supporting our conclusion that silencing RFC4 causes increase in K48-linked polyubiquitination of NICD1 proteins (new Supplementary Figure 6b). Corresponding data have been incorporated into the revised manuscript.

4. The CHX chase looks rather unimpressive in Fig 4d. CHX can interfere with the assay itself, and it should first be established that it doesn't by using a standard pulse chase assay.

Response: We thank the reviewer for raising this interesting suggestion. In this revision, we evaluated the effect of overexpressing or silencing RFC4 on turnover of exogenous NICD1 proteins by using a standard pulse chase assay. As described in Supplementary Methods in the revised manuscript, indicated cells were washed with pre-warmed PBS (37°C) to remove residual unlabeled amino acids, refreshed by pre-warmed MEM, supplemented with arginine, leucine, glucose, inositol, 0.2% BSA, and 10 mM HEPES (pH 7.4) and cultured for 15 min at 37°C with 5% CO₂ in a water-saturated atmosphere. Cell cultures were added with 4.3 MBq TRAN ³⁵S-LABEL for 30 min, washed with pre-warmed PBS and refreshed with standard culture medium supplemented with 5% FBS. Cells were harvested by RIPA lysis buffer at the indicated time points and immunoprecipitated by antibodies against NICD1. The immunoprecipitated fraction was separated on a standard discontinuous SDS gel. The gel was sequentially incubated in fixation buffer (40% methanol, 10% acetic acid, 50% distilled water), in H₂O, and in 1 M sodium salicylate each for 30 min at room temperature. The gel was put on 3MM paper,

covered with Saran wrap and dried at 70°C in a vacuum dryer until it is completely free of water. Finally, the radioactive proteins were detected by phosphorimager. As shown in new Supplementary Figure 6c, a half of 35S-labeled NICD1 proteins undergone degradation in less than 4 hours in vector-control A549 cells, whereas less than 20% of NICD1 proteins were degraded in 8 hours in RFC4-overexpressing A549 cells; by contrast, more than a half of NICD1 proteins were degraded in 1 hour in H1975 cells silenced with RFC4, whereas in vector-control H1975 cells only approximately a quarter of NICD1 proteins were degraded in 8 hours. These data further validate the potent promoting effect of RFC4 on stabilizing NICD1. Corresponding data have been incorporated into the revised manuscript on Page 11.

5. The IPs in Fig. 5b and 5c are completely uncontrolled. Even though IgG lanes are widely used as controls, I much prefer bait knock-downs, and the authors should use knock-downs instead in 5b and leave out the bait in 5c to demonstrate specificity of their interactions. They could also co-express both proteins in cells and do the IPs without using recombinant protein, along with the according controls.

Response: The reviewer's suggestion is well taken. In this revision, knock-down of RFC4 or Notch1 was used to confirm the specific interaction between RFC4 and NICD1. As shown in new Figure 5b, the IP assays immunoprecipitated with anti-NICD1 or anti-RFC4 antibodies revealed that the interaction between RFC4 and NICD1 was diminished when RFC4 and Notch1 was separately silenced. In parallel, when HA-tagged NICD1 was co-expressed with Flag-tagged RFC4, Flag-tagged GAPDH or the empty vector, the IP assays immunoprecipitated with anti-HA affinity agarose showed that only Flag-tagged RFC4 was pulled down; similarly, when Flag-tagged RFC4 was co-expressed with HA-tagged NICD1, HA-tagged p84 or the empty vector, the IP assays immunoprecipitated with anti-Flag affinity agarose showed that only HA-tagged NICD1 was pulled down (new Supplementary Figure 7c). These data further demonstrate specificity of the interaction between RFC4 and NICD1. Corresponding data have been incorporated into the revised manuscript on Page 12.

6. Fig 5f is insufficient to make the point of competitive binding between CDK8 and RFC4 to NICD. No cyclin was used in this assay, which is known to greatly alter the

conformation of the kinase. And how about the other way around? Do increasing amounts of CDK8 displace RFC4? Best is to examine the NICD - RFC4 interaction as in 5b and knock-down CDK8, or examine the NICD - CDK8 interaction with RFC4 knock-down.

Response: The authors thank the reviewer for raising the important advice. The surface plasmon resonance (SPR) kinetic analysis showed that the binding affinity between RFC4 and NICD1 was approximately five-folds higher than that between CDK8 and NICD1 (new Figure 5c). In consistence, similar to CDK8, the amounts of Cyclin C, which is a binding partner for CDK8, were also gradually decreased and even totally vanished in the pulled down proteins when HA-tagged NICD1 proteins were immunoprecipitated following addition of purified RFC4 in a dose-dependent manner (new Figure 5d). Interestingly, although the binding of NICD1 to RFC4 could be impaired by the increasing amounts of purified CDK8 in a dose-dependent manner, it appears that large amounts of CDK8 proteins are required to significantly abrogate the interaction between NICD1 and RFC4 (new Figure 5d). Moreover, while silencing RFC4 mildly enhanced the interaction between NICD1 and CDK8, silencing CDK8 significantly enhanced the interaction between NICD1 and RFC4 (new Figure 5e). In parallel, overexpressing RFC4 significantly abrogated the binding of NICD1 to CDK8 whereas CDK8 overexpression moderately impaired the binding of NICD1 to RFC4 (new Figure 5e). It is also important to note that from the TCGA lung cancer datasets the mRNA levels of both CDK8 and Cyclin C are slightly increased in NSCLC tissue as compared to normal lung tissue; by contrast, both mRNA and protein levels of RFC4 are significantly upregulated in NSCLC tissue as compared to normal lung tissue (new Supplementary Figure 9e; Supplementary Figure 2b for Reviewer 2). Indeed, CDK8 protein levels were rarely upregulated in 8 pairs of NSCLC tissue, where RFC4 protein levels were significantly upregulated in each pair of NSCLC tissue as compared to adjacent normal lung tissue (new Supplementary Figure 5a). These data suggest that both higher binding affinity of RFC4 to NICD1 and increased expression levels of RFC4 in NSCLC tissue make RFC4 more competitive than CDK8 to bind to NICD1, leading to abrogated phosphorylation and ubiquitination of NICD1 and thus stabilization of NICD1 proteins. Corresponding data have been incorporated into the revised manuscript on Pages 12 and 13.

7. The NICD is full of potential serine phosphorylation sites. By using a general p-Ser but otherwise not site-specific antibody, the authors have no idea what they are looking at.

Sure, the site seems CDK8 responsive (Fig 5h & i). But the relevant site in the Notch degron is actually a threonine (T2512). The contributing effects of S2514 and S2517 are controversial at best. Without knowing the site in question, there is no evidence for a mechanism.

Response: The authors thank the reviewer for raising these important comments. Since there are to date no available antibodies recognizing specific phosphorylation sites of NICD1, we used several distinct HA-tagged NICD1 mutants such as NICD1(S2514A), NICD1(S2517A), NICD1(S2514A/2517A), NICD1(T2512A), NICD1(T2542A) and NICD1(T2512A/2542A), which should be resistant to phosphorylation at these serine or threonine sites, to further investigate the effect of RFC4 on NICD1 phosphorylation. As shown in new Figure 5, when single phosphorylation of S2514 or S2517 was deprived by separately mutating serine to alanine, overexpressing RFC4 still apparently reduced total serine and threonine phosphorylation of NICD1(S2514A) or NICD1(S2517A), whereas RFC4 hardly decreased total serine and threonine phosphorylation of NICD1(S2514A/2517A) (new Supplementary Figure 7g). Interestingly, RFC4 significantly reduced total serine phosphorylation of NICD1(T2512A), NICD1(T2542A) or NICD1(T2512A/2542A), and only reduced total threonine phosphorylation of NICD1(T2512A) or NICD1(T2542A), but not of NICD1(T2512A/2542A) (new Supplementary Figure 7g). Notably, overexpressing RFC4 could remarkably abrogated the binding of NICD1 to MEKK1 or GSK3 β (new Supplementary Figure 7h), which are potential kinases able to phosphorylate NICD1 on T2512 and T2542, respectively ^{1,2}. Based on these data, we guess that the binding of RFC4 to the PEST domain of NICD1 might squeeze away other NICD1-interactive proteins, including various NICD1 kinases, probably due to space occupying effect, resulting in abrogation of NICD1 phosphorylation at multiple serine or threonine amino acids, such as S2514, S2517, T2512, and T2542, in the PEST domain. Of note, it appears that simultaneous phosphorylation at S2514 and S2517 is important for phosphorylation of NICD1 at T2512 and T2542 whereas the status of phosphorylation at T2512 or T2542 might not impact that of S2514 or S2517 phosphorylation. Indeed, we also found that the total threonine phosphorylation levels of NICD1(S2514A/2517A) are greatly decreased as compared to those of wild-type NICD1, whereas the total serine phosphorylation levels of NICD1(T2512A/2542A) are similar to those of wild-type NICD1 (Supplementary Figure 4b for Reviewer 4). However, due to lack of phosphorylation site-specific antibodies, it is still hard to draw a clear conclusion.

Appropriate data have been incorporated into the revised manuscript on Pages 13 and 14.

8. The point of Fig 5k is to demonstrate that enforced RFC4 expression displaces CDK8 from Notch complexes, which then would abolish degron phosphorylation and therefore diminish Fbxw7 binding. We have extensive experience with Fbxw7 in my lab. I therefore find these data unbelievable. For one, Fbxw7 protein is notoriously hard to detect by itself, let alone in co-IPs. But more importantly: without uncoupling of binding from turnover, a ubiquitin ligase, like Fbxw7, cannot be co-immunoprecipitated by a substrate as it would become degraded upon contact. Now, there are some exceptions. But Notch isn't one of them. In addition, it is our experience that the antibody used in this study (3A9) makes an Fbxw7-independent cross-reaction right underneath where untagged Fbxw7 would migrate on a gel. Therefore, this experiment needs to be better controlled using Fbxw7 knock-downs, or better crispr-mediated knockout. There are also cell lines available with targeted deletion of Fbxw7.

Response: The authors greatly thank the reviewer for providing above valuable information and raising important suggestion. Firstly, we are sorry for providing misleading information in the original methods and in fact we used the 3A9 antibody (ab74054, Abcam) against FBXW7 to detect Notch1-interactive FBXW7 in the immunoprecipitated fraction pulled down by Notch1 of the IP assay and used another antibody ab109617 (Abcam) against FBXW7 to detect FBXW7 in the input fraction of the IP assay, for which the proteasome inhibitor MG132 was added for 6 hours prior to the experiment. In this revision, when the similar IP assay was performed in A549 cells with FBXW7 knocked out by the Crispr-Cas9 system, the 3A9 antibody against FBXW7 still detected strong immunoblot bands similar to those in vector-control A549 cells, indicating that the previous strong immunoblot bands of FBXW7 detected by 3A9 in original Figure 5k are indeed non-specific (Supplementary Figure 4a for Reviewer 4). However, the other antibody ab109617 could detect immunoblot bands of FBXW7 immunoprecipitated by Notch1 in the vector-control A549 cells, but not in FBXW7-depleted A549 cells, both of which were pre-treated with MG132; thus, we proved that overexpressing RFC4 is indeed able to abrogate the interaction between FBXW7 and NICD1 (Figure 5, h and i). In consistence, silencing RFC4 failed to cause K48-linked polyubiquitination of NICD1 or decrease NICD1 levels in FBXW7-silenced or FBXW7-depleted A549 cells (Figure 5, j and k). Taken together, these data further demonstrate that enforced RFC4 expression

displaces CDK8 from Notch, which abolishes degron phosphorylation and diminishes FBXW7 binding to NICD1, leading to NICD1 stabilization. Appropriate data have been incorporated into the revised manuscript on Page 14.

9. For Fig 5l, see my comments in 3. You could literally look at Fbxw7 auto-ubiquitylation, but sell it for Notch.

Response: The authors thank the reviewer for this advice. In consistence with the results obtained by the IP assays immunoprecipitated with Flag-tagged NICD1 (new Figure 5j presented as original Figure 5l in the initial version of manuscript), the IP assays immunoprecipitated with anti-HA affinity agarose showed that the binding of HA-tagged K48-linked ubiquitin (Ubk48-HA) to Flag-tagged NICD1 (NICD1-Flag) was greatly impaired when CDK8 or FBXW7 was silenced, whereas silencing RFC4 failed to reverse the binding of HA-tagged K48-linked ubiquitin to Flag-tagged NICD1 when CDK8 or FBXW7 was pre-silenced (new Supplementary Figure 7i), further supporting the conclusion that RFC4 abrogates CDK8/FBXW7-mediated polyubiquitination of NICD1. Corresponding data have been incorporated into the revised manuscript on Page 14.

Overall, this is good work. The functional link of NICD and RFC4 is well illuminated. But the mechanism is still rather unclear.

Response: The authors greatly appreciate the reviewer's inspiring comments and very constructive suggestion.

References

1. Lee HJ, Kim MY, Park HS. Phosphorylation-dependent regulation of Notch1 signaling: the fulcrum of Notch1 signaling. *BMB Rep* **48**, 431-437 (2015).
2. Morrugares R, *et al.* Phosphorylation-dependent regulation of the NOTCH1 intracellular domain by dual-specificity tyrosine-regulated kinase 2. *Cell Mol Life Sci* **77**, 2621-2639 (2020).

Supplementary Figure 4 for Reviewer 4

a

b

REVIEWERS' COMMENTS

Reviewer #2 (Remarks to the Author):

The authors have adequately addressed the main issues raised during the first review process. The mechanistic data presented regarding the interaction of RFC4 and notch signalling provide strong evidence for a novel link between RFC4 expression and Notch signalling that will be of interest in the field - but the relationship between RFC4 expression and outcomes in NSCLC remain correlative and therefore need to be interpreted cautiously, in particular as RFC4 expression correlates with cell proliferation markers.

Specific points

line 72. The response rates of patients with metastatic NSCLC are not 'extremely low' Provide more specific information

lines 103/104 The text 'which almost resulted in therapeutic failures' does not make sense

lines 168-175 The terms 'High' and 'Low' metastatic lines (for A549 and H1975) are not defined

Comments: In NSCLC, the RFC4 gene is commonly amplified in squamous cell but not adenocarcinoma (TCGA), but this is not discussed, and the human cell lines used in the study are adenocarcinoma.

The specificity of these observations to NSCLC are not clear as (e.g.) esophageal and ovarian cancers also have a high proportion of RFC4 gene amplification, and mRNA expression is highly correlated with markers of cell proliferation (TCGA) for these diseases too - these limitations should be addressed in the discussion as the data could (potentially wrongly) indicate selective effects for NSCLC, in particular as the most common NSCLC histological subtype (adenocarcinoma) rarely has RFC4 gene amplification

Reviewer #3 (Remarks to the Author):

The authors have carefully addressed and extensively revised the manuscript in response to the reviewers' comments and added a substantial amount of new data in support of their findings.

Two minor comments: LLC is not really a lung cancer cell line, but a sarcoma, and the Figure 5 legend has "FLAG" misspelled in the first sentence as "FALG".

Reviewer #4 (Remarks to the Author):

The authors have addressed all my points of concern adequately. My last comment concerns the migration of Fbxw7 on SDS-PAGE. In figure 5 as well as supplemental figure 7 the migration of Fbxw7 is marked at just under 70 kDa and in my reviewers figure 4 Fbxw7 migrates just above 40 kDa. The authors should revisit their migration pattern as the most common Fbxw7 alpha isoform is established to migrate at around 110 kDa (~30 kDa larger than calculated). There are indeed 2 smaller isoforms, which would migrate around the marked 70 kDa range (Fbxw7 beta is about 69 kDa). But to my knowledge this isoform is much harder to detect than the alpha isoform, and it is cytoplasmic.

REVIEWERS' COMMENTS

Reviewer #2 (Remarks to the Author):

The authors have adequately addressed the main issues raised during the first review process.

The mechanistic data presented regarding the interaction of RFC4 and notch signalling provide strong evidence for a novel link between RFC4 expression and Notch signalling that will be of interest in the field - but the relationship between RFC4 expression and outcomes in NSCLC remain correlative and therefore need to be interpreted cautiously, in particular as RFC4 expression correlates with cell proliferation markers.

Response: The authors greatly appreciate the reviewer's encouraging and constructive comments. Our data strongly suggest that through activating Notch signaling and forming a positive feedback loop with NICD1, RFC4 potently promotes tumor invasion/metastasis and stemness/tumorigenicity, which can be independent of its role in DNA replication and repair and of its ability to enhance cell proliferation/cell cycle/survival. Indeed, as RFC4 expression significantly correlates with cell proliferation markers in human NSCLC tissue, the oncogenic effects of RFC4 on cell proliferation, tumorigenicity and tumor metastasis could together contribute to the worse outcomes of NSCLC patient prognosis. To ease the reviewer's concern, in this revision we have modified the interpretation of the roles of RFC4 in NSCLC appropriately and appropriate description has also been incorporated into the Discussion Section in the revised manuscript on Page 18.

Specific points

line 72. The response rates of patients with metastatic NSCLC are not 'extremely low'

Provide more specific information

Response: We thank the reviewer for pointing out the inappropriate description. The response rates of patients with metastatic NSCLC to first-line chemotherapy, targeted therapy and immunotherapy can be ranging from 15%-30%, 25%-60%, and 14%-65%, respectively^{9, 10, 11}. As response rates might be impacted by selected therapy types, prescribed medication, enrolled patients and so on, it is hard to give a definitive calculation of the response rates of metastatic NSCLC patients to clinical therapies. Thus, we prefer to change the original description to "Despite the significant advancement in currently

available therapies, the unsatisfying response rates of metastatic NSCLC patients to the initial anti-cancer treatments and the fairly high frequencies of tumor recurrence posttreatment remain to be the most serious challenge in the clinic.”

lines 103/104 The text 'which almost resulted in therapeutic failures' does not make sense

Response: In this revision, “which almost resulted in therapeutic failures” has been changed into “which almost ended in therapeutic failures”.

lines 168-175 The terms 'High' and 'Low' metastatic lines (for A549 and H1975) are not defined

Response: According to a recent paper, A549 cell line is moderate metastatic, while H1975 cell line presents highly metastatic potential (*Nature*. 2020 Dec;588(7837):331-336.). To avoid misunderstanding, in this revision, we have deleted these terms and made appropriate modification according to the context of this paragraph on page 7.

Comments: In NSCLC, the RFC4 gene is commonly amplified in squamous cell but not adenocarcinoma (TCGA), but this is not discussed, and the human cell lines used in the study are adenocarcinoma.

The specificity of these observations to NSCLC are nor clear as (e.g.) esophageal and ovarian cancers also have a high proportion of RFC4 gene amplification, and mRNA expression is highly correlated with markers of cell proliferation (TCGA) for these diseases too - these limitations should be addressed in the discussion as the data could (potentially wrongly) indicate selective effects for NSCLC, in particular as the most common NSCLC histological subtype (adenocarcinoma) rarely has RCF4 gene amplification.

Response: We thank the reviewer for this important suggestion. In fact, we have employed a LUSC cell line (H1703) to prove the pro-invasive and pro-tumorigenic effects of RFC4 and RFC4-directed stabilization of NCD1 proteins and thus activation of Notch signaling (Fig. 3a, 6f, 6g, S1 a-d, S3 e-h, S4 a-g and S8 a-c, e and f), which are consistent with the results obtained from multiple LUAD cell lines. Of note, amplification of the 3q chromosome, where the *RFC4* gene is located, is mostly seen in squamous cancers (such as LUSC, esophageal and ovarian cancers). Consistently, the mRNA levels of RFC4 are averagely increased approximately 9.42-fold in LUSC, approximately 40.3% of which have *RFC4* gene amplification in the TCGA lung cancer datasets; by contrast, RFC4 mRNA levels are increased approximately 3-fold on average in LUAD, and approximately 2.33% of LUAD samples have *RFC4* gene amplification in the TCGA lung cancer datasets. Our

study suggests that both the transcriptional upregulation of RFC4 by activated Notch1 signaling and *RFC4* amplification should contribute to high levels of RFC4 to varying degrees in LUAD and LUSC, both of which could utilize the pivotal roles of the positive feedback loop between RFC4 and NICD1 in coupling NSCLC metastasis and stemness properties. In addition, other cancer types, such as esophageal and ovarian cancers, in which aberrantly activated Notch1 signaling plays important roles, also have distinct high proportions of *RFC4* gene amplification, and high RFC4 levels significantly correlates with poor prognosis of patients with these cancers, indicating that the oncogenic effects of RFC4 are not limited to NSCLC. Appropriate information has been incorporated into the Discussion Section in the revised manuscript on Pages 19 and 20.

Reviewer #3 (Remarks to the Author):

The authors have carefully addressed and extensively revised the manuscript in response to the reviewers' comments and added a substantial amount of new data in support of their findings.

Two minor comments: LLC is not really a lung cancer cell line, but a sarcoma, and the Figure 5 legend has "FLAG" misspelled in the first sentence as "FALG".

Response: The authors are greatly thankful for the reviewer's encouraging comment. It is quite interesting and important to know that the LLC cell line is more like a sarcoma. Our data suggest that the pro-invasive and pro-tumorigenic effects of RFC4 through stabilizing NICD1 and overactivating the Notch1 signaling might not be selective effects for lung cancer. However, the Lewis lung carcinoma (LLC, or named as LL/2) cell line is documented in ATCC as a cell line established from the lung of a C57BL mouse bearing a tumor resulting from an implantation of primary Lewis lung carcinoma and has been widely used as lung cancer tumorigenesis and metastasis cell models (*Nat Cell Biol.* 2021 Feb;23(2):172-183.; *Nat Microbiol.* 2021 Mar;6(3):277-288.; *Mol Cell.* 2021 Feb 13;S1097-2765(21)00010-1.; *Cell Metab.* 2018 Aug 7;28(2):243-255.). To avoid misleading description, we have made appropriate modification about the information of LLC on Pages 7 and 21. We prefer to present the data obtained from LLC in this revised manuscript and are also willing to remove them if the reviewer/editor feels necessary. In addition, the misspelled "FALG" in the Figure 5 legend has been corrected to "FLAG".

Reviewer #4 (Remarks to the Author):

The authors have addressed all my points of concern adequately. My last comment concerns the migration of Fbxw7 on SDS-PAGE. In figure 5 as well as supplemental figure 7 the migration of Fbxw7 is marked at just under 70 kDa and in my reviewers figure 4 Fbxw7 migrates just above 40 kDa. The authors should revisit their migration pattern as the most common Fbxw7 alpha isoform is established to migrate at around 110 kDa (~30 kDa larger than calculated). There are indeed 2 smaller isoforms, which would migrate around the marked 70 kDa range (Fbxw7 beta is about 69 kDa). But to my knowledge this isoform is much harder to detect than the alpha isoform, and it is cytoplasmic.

Response: We greatly appreciate the reviewer for the encouraging comment and pointing out the issue. The authors have reviewed the original Western blot images of FBXW7 detected by the ab109617 (Abcam) antibody against FBXW7, and noticed that the protein markers of FBXW7 immunoblots in our submitted figures were indeed mislabeled below 70kDa, which should be at around 110kDa. Has not it been reminded by the reviewer, the authors had been confusing about the 110kDa migration pattern of FBXW7 as detected by the ab109617 antibody in the Western blot assays and had thought that there might be something wrong with the prestained protein marker. We are very sorry for this inappropriate mistake. To further confirm the migration pattern of FBXW7, a plasmid containing Flag-tagged FBXW7 α (NM_033632, the longest isoform of FBXW7) was transfected into A549 and 293FT cells. Western blot assays showed that both the FBXW7 antibody and Flag antibody detected the immunoblot bands of FBXW7 around 110 kDa (new Supplementary Figure 4c for Reviewer 4). Notably, the main isoform of endogenous FBXW7 proteins in several NSCLC cell lines, primary NSCLC cells (LC1) and 293FT cells, LLC cells and NSCLC cell lines migrated at around 110 kDa, while it appears that smaller isoforms of FBXW7 migrating at around 70 kDa could also be faintly detected (new Supplementary Figure 4d for Reviewer 4). In addition, we also would like to apologize for misspelling the protein marker in original Supplementary Figure 4a for Reviewer 4, in which the migration of FBXW7 is between 100 kDa and 130 kDa (new Supplementary Figure 4a for Reviewer 4). Corresponding correction has been made for labeling the protein markers of FBXW7 immunoblot bands in our submitted figures between 100kDa and 130kDa.

Supplementary Figure 4 for Reviewer 4

References

1. Stransky N, *et al.* The Mutational Landscape of Head and Neck Squamous Cell Carcinoma. *Science* **333**, 1157-1160 (2011).
2. Agrawal N, *et al.* Comparative Genomic Analysis of Esophageal Adenocarcinoma and Squamous Cell Carcinoma. *Cancer Discovery* **2**, 899-905 (2012).
3. Egloff AM, Grandis JR. Molecular Pathways: Context-Dependent Approaches to Notch Targeting as Cancer Therapy. *Clin Cancer Res* **18**, 5188-5195 (2012).
4. Westhoff B, *et al.* Alterations of the Notch pathway in lung cancer. *P Natl Acad Sci USA* **106**, 22293-22298 (2009).
5. Zou B, Zhou XL, Lai SQ, Liu JC. Notch signaling and non-small cell lung cancer (Review). *Oncol Lett* **15**, 3415-3421 (2018).
6. Wang NJ, *et al.* Loss-of-function mutations in Notch receptors in cutaneous and lung squamous cell carcinoma. *P Natl Acad Sci USA* **108**, 17761-17766 (2011).
7. Lee HJ, Kim MY, Park HS. Phosphorylation-dependent regulation of Notch1 signaling: the fulcrum of Notch1 signaling. *BMB Rep* **48**, 431-437 (2015).
8. Morrugares R, *et al.* Phosphorylation-dependent regulation of the NOTCH1 intracellular domain by dual-specificity tyrosine-regulated kinase 2. *Cell Mol Life Sci* **77**, 2621-2639 (2020).
9. Taniguchi Y, *et al.* Impact of metastatic status on the prognosis of EGFR mutation-positive non-small cell lung cancer patients treated with first-generation EGFR-tyrosine kinase inhibitors. *Oncol Lett* **14**, 7589-7596 (2017).
10. Schvartsman G, *et al.* Response rates to single-agent chemotherapy after exposure to immune checkpoint inhibitors in advanced non-small cell lung cancer. *Lung Cancer* **112**, 90-95 (2017).
11. Herbst RS, *et al.* Atezolizumab for First-Line Treatment of PD-L1-Selected Patients with NSCLC. *N Engl J Med* **383**, 1328-1339 (2020).